# STaR-Bets: Sequential Target-Recalculating Bets for Tighter Confidence Intervals

**Václav Voráček**
Second Foundation*
vasek.voracek@gmail.com

**Francesco Orabona**
King Abdullah University of Science and Technology
francesco@orabona.com

## Abstract

The construction of confidence intervals for the mean of a bounded random variable is a classical problem in statistics with numerous applications in machine learning and virtually all scientific fields. In particular, obtaining the tightest possible confidence intervals is vital every time the sampling of the random variables is expensive. The current state-of-the-art method to construct confidence intervals is by using betting algorithms. This is a very successful approach for deriving optimal confidence sequences, even matching the rate of law of iterated logarithms. However, in the fixed horizon setting, these approaches are either sub-optimal or based on heuristic solutions with strong empirical performance but without a finite-time guarantee. Hence, no betting-based algorithm guaranteeing the optimal $\mathcal{O}(\sqrt{\frac{\sigma^2 \log \frac{1}{\delta}}{n}})$ width of the confidence intervals are known. This work bridges this gap. We propose a betting-based algorithm to compute confidence intervals that empirically outperforms the competitors. Our betting strategy uses the optimal strategy in every step (in a certain sense), whereas the standard betting methods choose a constant strategy in advance. Leveraging this fact results in strict improvements even for classical concentration inequalities, such as the ones of Hoeffding or Bernstein. Moreover, we also prove that the width of our confidence intervals is optimal up to an $1 + o(1)$ factor diminishing with $n$. The code is available on github.

## 1 Introduction

Quantifying uncertainty is a cornerstone of statistical inference, and Confidence Intervals (CIs) remain one of the most widely used tools for this purpose. Typically constructed within the frequentist paradigm, a $(1 - \delta)$-CI provides a range of plausible values for an unknown parameter, derived from observed data, with the guarantee that the procedure yields intervals covering the true parameter value in $1 - \delta$ proportion of hypothetical repetitions. In particular, in this paper we are interested in providing CI for the mean of a bounded random variable. So, more formally, after observing $n$ i.i.d. samples $X_1, \ldots, X_n$ of a random variable in $[0, 1]$ with mean $\mu$, we want to find $l_n$ and $u_n$ such that

$$\mathbb{P}\{\mu \leqslant u_n\} \geqslant 1 - \delta/2 \ \text{ and } \ \mathbb{P}\{\mu \geqslant l_n\} \geqslant 1 - \delta/2 \ . \tag{1}$$

We aim for narrow confidence intervals, i.e., small $u_n - l_n$. Classical methods provide well-established procedures with optimal asymptotic properties, but their performance deteriorates when $n$ is small.

This paper explores an alternative approach to constructing confidence intervals, based on the framework of game-theoretic probability and sequential betting. In this framework, the algorithm makes sequential bets on the values of $X_i - m$. If $m$ is close to the true mean $\mu$, then the random

---

*work done while with University of Tübingen.

39th Conference on Neural Information Processing Systems (NeurIPS 2025).

variables $X_i - m$ is approximately zero mean. Now, it is intuitive that no strategy can hope to gain money betting on a fair random variable and we expect little money betting on a random variable with mean very close to zero. Conversely, if the algorithm makes enough money, we can infer that $m$ is far from the true mean $\mu$. The above reasoning can be carried out in rigorous way, using the concept of *test martingales* [20]. In particular, one can show that we can construct the confidence interval as the set of all values of $m$ such that the wealth does not reach $\frac{2}{\delta}$. In this view, a better betting algorithm results in a tighter confidence interval.

These approaches give state-of-the-art theoretical and empirical results in the time-uniform case. However, they fall short in the finite-sample regime, both in theory and in practice.

**Contributions.** In this paper, we propose a new strategy for betting in the finite-sample regime. We explicitly take into account how many rounds we still have and that by how much we still need to multiply our current wealth to reach the threshold. This give rise to a new family of betting algorithms for the finite-sample setting, the ✴-algorithms (STaR=Sequential Target-Recalculating). We show that this strategy can be used, for example, to immediately improve the confidence intervals calculated through Hoeffding and Bernstein, *without losing anything*. Moreover, we present a new algorithm, STaR-Bets, that achieve valid state-of-the-art confidence intervals on a variety of distributions. Additionally, we show that its variant, Bets, attains the optimal rates for confidence interval width—marking the first such result for betting-based confidence interval methods.

## 2 Related Work

The literature on how to construct confidence intervals for the mean of a random variable is vast. Classic results like Hoeffding's [10] and Bernstein's [3] inequalities require the knowledge of the variance of the random variable (which can always upper bounded for bounded random variables). Later, Maurer and Pontil [12] and Audibert et al. [2] proved that it is possible to prove *empirical* Bernstein's inequalities, that is, with optimal dependency on the variance but without its prior knowledge. However, these results tend to be loose on small samples because they sacrifice the tightness of the interval with a simple and interpretable formula.

On the other hand, it is possible to design procedures to numerically find the intervals, without having a closed formula. A classic example is the Clopper-Pearson confidence interval [5] for Bernoulli random variables, that is obtained by inverting the Cumulative Distribution Function (CDF) of the binomial distribution. This approach is general, in the sense that it can be applied to any random variable when the CDF is known. However, no optimal solution is known if we do not have prior knowledge of the CDF of the random variable.

An alternative approach is the betting one, proposed by Cover [6] for the finite-sample case and by Shafer and Vovk [19] and Shafer et al. [20] for the time-uniform case, as a general way to perform statistical testing of hypotheses. The first paper to consider an implementable strategy for testing through betting is Hendriks [9]. Later, Jun and Orabona [11] proposed to design the betting algorithms by specifically minimizing the regret w.r.t. a constant betting scheme. Waudby-Smith and Ramdas [23] suggested the use of betting heuristics, motivated by asymptotic analyses and online convex optimization approaches to betting algorithms [14, 8]. Recently, Orabona and Jun [13] have shown that regret-based betting algorithms based on universal portfolio algorithms [7] can recover the optimal law of iterated logarithm too, and achieve state-of-the-art performance for time-uniform confidence intervals. However, these approaches fail to give optimal widths in the finite-sample regime that we consider in this paper. We explicitly note that even the optimal portfolio strategy for the finite-time setting [15] does not achieve the optimal widths because it suffers from an additional $\log T$ factor. This is due to the fact that minimizing the regret with respect to a fixed strategy does not seem to be right objective to derive confidence intervals in the finite-sample case.

The problem of achieving the tightest possible confidence interval for random variables for small samples has received a lot of interest in the statistical community. It is enough to point out that even the Clopper-Pearson approach has been considered to be too wasteful, because of the discrete nature of the CDF that does not allow an exact inversion.[2] In this view, sometimes people prefer to use approximate methods, that is, methods that do not guarantee the exact level of confidence $1 - \delta$, because they are less conservative [see, e.g., the reviews in 1, 18]. Recently, Phan et al. [16] proposed an approach based on constructing an envelope that contains the CDF of the random variable with

---

[2]See Voracek [22], for an optimal, randomized procedure.

high probability. This approach was very promising in the small sample regime, however in Section 6 we will show that our approach is consistently better.

In this paper, we build on the ideas of Cover [6] regarding the construction of optimal betting strategies for Bernoulli random variables and design a general betting scheme in that spirit.

## 3 A Coin-Betting Game

The statistical testing framework we rely on is based on *betting*. In particular, we will set up simple betting games based on the hypothesis we want to test. The outcome of the testing will be linked to how much money a betting algorithm makes. Hence, we will be interested in designing "optimal" betting strategies. In the following, we first describe a simple betting game to draw some intuition about the optimal betting strategies. Then, in Section 4 we formally introduce the testing by betting framework.

We introduce a simple sequential coin betting game in which we bet on the outcomes of a coin toss. If the result is Heads, we gain the staked amount, otherwise we lose it. The game is as follows: We will bet for $n$ rounds, and we win if we multiply our initial wealth by at least a factor of $k$. Hence, we are not interested in how much money we make, but only if we pass a certain threshold.

Let's now discuss some possible scenarios, to see how the optimal betting strategy changes.

**Scenario 1, $n = 1$, $k = 2$:** It is optimal to bet everything and if the outcome is Heads, we win. Generally, in the last round we should always bet everything we have.
**Scenario 2, $n = 5$, $k = 0.9$:** In this case, it is optimal to bet anything, and we win.
**Scenario 3, $n = 2$, $k = 4/3$:** Consider the possible outcomes: Heads/Heads, Heads/Tails, Tails/Heads, Tails/Tails. In the last case we always lose. If we want to win in the Heads/Tails case, we would need to bet $1/3$, in which case we reach the desired wealth after observing Heads and we stop betting. In the case we first observe Tails and we end up with $2/3$ money. Then, we recover Scenario 1 and bet all the money.

Before we continue with more general scenarios, we make some observations. These considerations will be useful when we will link betting strategies to the proof of standard concentration inequalities, such as Hoeffding's or Bernstein's.

- The original values of $n, k$ do not matter. Instead, what matter are how these quantities "change over time," that is, how many rounds are left and how many times we still need to multiply our wealth.

- When we hit the required wealth, we should stop betting.

- For a given $n$, the smaller $k$ is, the less we need to bet. Conversely, if $k$ is large, we need to bet aggressively.

- We should always finish the game by either going bankrupt or by hitting the target.

Already in [6, Example 3], it was shown that if a betting strategy is optimal (for hypothesis testing, in the sense of Proposition 7), it either has to reach the target, or go bankrupt. The contrary was also proven, that there exists such an algorithm. More generally, all possible outcomes of the coin-betting game were described. Let's now consider more complex scenarios.

**Scenario 4, $n, k \approx 1 + 2^{-n}$:** In this case, our bets would be $\approx 2^{-n}$, so that we win as soon as we observe a single heads outcome.
**Scenario 5, $n, k \approx 2^n$:** In this case, our bets would be $\approx 1$ in general, as we need to roughly double our wealth every round and hope for lots of heads.
**Scenario 6, $n, k$:** In the general case, we bet $\approx \sqrt{\log k/n}$. In this way, if we observe $H$ heads and $T$ tails, the wealth would be roughly

$$\left(1 + \sqrt{\frac{\log k}{n}}\right)^H \left(1 - \sqrt{\frac{\log k}{n}}\right)^T \approx \left(1 - \frac{\log k}{n}\right)^T \left(1 + \sqrt{\frac{\log k}{n}}\right)^{H-T} \gtrsim k,$$

as long as $H - T \gtrsim \sqrt{n \log k}$.

Let us examine the event $H - T \gtrsim \sqrt{n \log k}$: If the coin is fair, the event $H - T \gtrsim \sqrt{n \log k}$—in which case we successfully multiplied our wealth by a factor $k$—happens with probability at most $\approx \frac{1}{k}$, and it is not just a coincidence. Thus, if we multiply our wealth by factor of $k$, we can rule out the possibility that the game is fair at a confidence level $1 - \frac{1}{k}$. This statement is the cornerstone of the testing by betting framework and we will make it formal and prove it in the next section.

## 4 Testing and Confidence Intervals by Betting

Before describing this technique in detail, we introduce the precise mathematical framework. We start with the definition of test processes: Sequences of random variables modeling fair[3] betting games. In these games, our wealth is always non-negative and stays constant (or decreases) in expectation. In the literature, test processes are also called e-processes and they are (super)-martingales.

**Definition 1** (Test process). Let $W_0, W_1, \ldots W_n$ be a sequence of non-negative random variables. We call it a test process if $\mathbb{E}[W_0] = 1$ and $\mathbb{E}[W_{i+1} \mid W_0, \ldots, W_i] \leqslant W_i$, for $i \geqslant 0$.

Next, Markov's inequality quantifies how unlikely it is for a test process to grow large.

**Proposition 1** (Markov's inequality). *Let $W_0, W_1 \ldots, W_n$ be a test process. For any $\delta \in (0, 1)$ it holds that $\mathbb{P}\left\{W_n \geqslant \frac{1}{\delta}\right\} \leqslant \delta$.*

*Proof.* $\mathbb{E}[W_n] \leqslant \mathbb{E}[W_0] = 1$ and $W_n > 0$. Then, $\mathbf{1}_{W_n \geqslant \frac{1}{\delta}} \leqslant \delta W_n$ and take expectation. $\square$

Finally, we introduce the betting-based confidence intervals.

**Theorem 2.** *A confidence interval obtained within the following scheme has coverage at least $1 - \delta$.*

---

### Confidence interval by betting

| | |
|---|---|
| **Objective:** | Construct a $(1 - \delta)$-confidence interval for $\mathbb{E}[X]$ from $X_1, \ldots, X_n \in [0, 1]$. |
| **Procedure:** | For each $m \in [0, 1]$, form a null hypothesis $H_0(m) : \mathbb{E}[X] = m$. |
| | Let $S = \{m \mid H_0(m) \text{ not rejected by betting at } 1 - \delta \text{ level}\}$. |
| **Outcome:** | Interval $I$ such that $S \subset I$, then $I$ is a $(1 - \delta)$-CI for $\mathbb{E}[X]$. |

### Testing by betting

| | |
|---|---|
| **Objective:** | Test the null hypothesis $H_0$ on $X_1, \ldots, X_n$ at confidence level $1 - \delta$. |
| **Procedure:** | Construct $W_1, \ldots, W_n$ such that it is a test process under $H_0$. |
| **Outcome:** | If $W_n \geqslant \frac{1}{\delta}$, reject $H_0$ at confidence level $1 - \delta$. |

---

*Proof.* If the mean $\mu$ is not contained in the resulting confidence interval, the null hypothesis $H_0(\mu) : \mathbb{E}[X] = \mu$ was rejected by betting. The corresponding sequence $W_1^\mu, \ldots, W_n^\mu$ is a test process and because it was rejected, we have $W_n^\mu \geqslant \frac{1}{\delta}$. But this happens with probability at most $1 - \delta$ by Markov's inequality. $\square$

**Test process under a null hypothesis:** A popular process (optimal in a certain sense [4]) for testing a null hypothesis $H_0(m) : \mu = m$ for a given $m \in [0, 1]$ is:

$$W_0^m = 1 \qquad\qquad W_i^m = W_{i-1}^m \left(1 + \ell_i \cdot (X_i - m)\right), \text{ for } i \geqslant 1, \qquad (2)$$

where $\ell_i \in \left[\frac{1}{m-1}, \frac{1}{m}\right]$ is our betting strategy. The betting strategy is, of course, independent of $X_{\geqslant i}$ but can depend on $X_{<i}$ and the other known quantities $m, n, \delta$. Under the null hypothesis, we have $\mathbb{E}[W_{i+1} \mid W_1, \ldots W_i] = \mathbb{E}[W_i]$. Additionally, by the bound on $\ell_i$, we ensure that $W_i^m \geqslant 0$, and so $\{W_i^m\}_{i=1}^n$ is a test process. We have already used this test process in the coin-betting example, if we identify Heads (resp. Tails) with 1 (resp. 0) and set $m = \frac{1}{2}$, then $\ell_i/2$ is the fraction of money we bet.

Theorem 2 provides a scheme on how to construct confidence intervals by testing all the possible values $m$ of the expectation of the random variable. Such an approach would require us to test a continuum of hypotheses which is impossible in general. There are two standard ways to proceed:

---

[3]We allow for the game to be unfair against us.

| **Algorithm 1** Hoeffding testing | **Algorithm 2** �- Hoeffding testing |
|---|---|

**Algorithm 1** Hoeffding testing
**Require:** i.i.d. $X_1, \ldots, X_n \in [0, 1]$
**Require:** $\delta > 0, m \in [0, 1], n$
  $W \leftarrow 1$
  **for** $t = 1 \ldots n$ **do**
    $\ell^H \leftarrow \sqrt{8 \left( \log \frac{1}{\delta} \right) / n}$
    $W \leftarrow W \exp \left( \ell^H \cdot (X_t - m) - \frac{(\ell^H)^2}{8} \right)$
  **if** $W \geqslant \frac{1}{\delta}$ **then**
    Reject $H_0(m) : m = \mathbb{E}[X]$

**Algorithm 2** ✱- Hoeffding testing
**Require:** i.i.d. $X_1, \ldots, X_n \in [0, 1]$
**Require:** $\delta > 0, m \in [0, 1], n$
  $W \leftarrow 1$
  **for** $t = 1 \ldots n$ **do**
    $\ell^H_{\star} \leftarrow \sqrt{8 \left( \log \frac{1}{W\delta} \right)_+ / (n - t + 1)}$
    $W \leftarrow W \exp \left( \ell^H_{\star} \cdot (X_t - m) - \frac{(\ell^H_{\star})^2}{8} \right)$
  **if** $W \geqslant \frac{1}{\delta}$ **then**
    Reject $H_0(m) : m = \mathbb{E}[X]$

1. **Root finding:** If the function $m \mapsto W_n^m$ has a simple structure (e.g., quasi-convex), then the set of non-rejected mean candidates form an interval and we can easily find its end points by numerical methods, or even in a closed form.

2. **Discretization:** Discretize the interval $[0, 1]$ into $G$ grid points $m_1 < m_2 < \cdots < m_G$. Use the same betting strategies for all $m$ in $m_i \leqslant m \leqslant m_{i+1}$ and all $1 \leqslant i \leqslant G - 1$. In this case, the function $m \mapsto W_n^m$ is monotonic in between the grid points, and so evaluating $m \mapsto W_i^m$ only at grid-points provide sufficient information for the construction of the confidence interval. We adopt this approach.

## 4.1 Hoeffding and Sequential Target-Recalculating (✱) Bets

In this section, we recover the classical confidence intervals by Hoeffding's inequality within the betting framework using Theorem 2—recovering Bernstein's intervals follows the same steps, so it is deferred to Appendix B. Our plan is to show that the Hoeffding betting strategy violates the design principles we have discussed earlier in this section. Hence, we can fix those problems and obtain tighter confidence intervals, essentially for free. More generally, we introduce ✱-betting strategies as the ones that have the better design principles.

We will use the following lemma to derive the test processes.

**Lemma 3** (Hoeffding's lemma). *Let $X \in [0, 1]$ with mean $\mu = \mathbb{E}[X]$, then*

$$\mathbb{E} \left[ e^{\ell \cdot (X - \mu) - \frac{\ell^2}{8}} \right] \leqslant 1, \quad \forall \ell \in \mathbb{R} . \tag{3}$$

*Furthermore, the sequence $W_0 = 1$, $W_{i+1} = W_i \exp \left( \ell \cdot (X_i - \mu) - \frac{\ell^2}{8} \right)$ for $i \geqslant 0$ is a test process.*

We compare the testing-by-betting algorithms. The standard Hoeffding test in Algorithm 1, and our improved, ✱-Hoeffding test in Algorithm 2. We observe that the constant betting of Algorithm 1 violates our desiderata for a good betting algorithm. Concretely, it may happen that at some point $W \geqslant \frac{1}{\delta}$, but we keep betting and end up with $W \leqslant \frac{1}{\delta}$. Additionally, it does not adapt to the current situation of how much do we need to increase $W$ and in how many rounds. In particular, the terms $n$ and $\log \frac{1}{\delta}$ defining the bet become irrelevant as $t$ increases. Instead, as we said, the only important quantities are i) $\frac{1}{\delta W}$: how many times we need to multiply our wealth and ii) $n - t$: the number of remaining rounds. This motivates the following definition.

> We call a betting algorithm for a test process $\{W_t\}_{t=1}^n$ a ✱-algorithm if it uses the quantities the $\log \frac{1}{W_t \delta}$ and $n - t$ to compute the bet at time $t$.

As an example, we can immediately generate the ✱-version of the Hoeffding algorithm in Algorithm 2 and prove that it is never worse than the original algorithm.

**Proposition 4.** *Let $m \in [0, 1]$. Whenever Algorithm 1 rejects the null hypothesis $H_0(m) : m = \mathbb{E}[X]$, then so does Algorithm 2 if they share the realizations $X_t$, $1 \leqslant t \leqslant n$.*

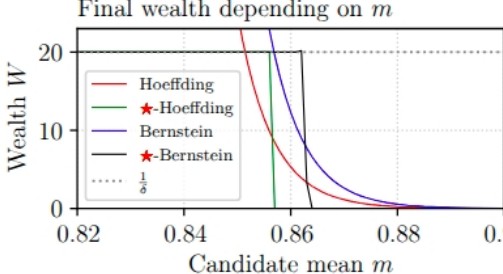
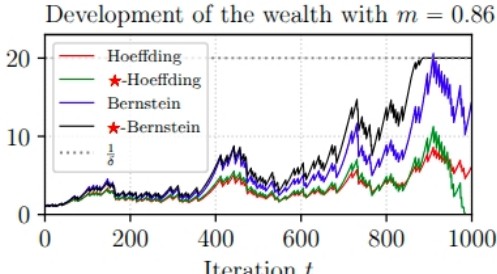

Figure 1: Comparison of the Algorithms 1,2, 5, and 6 with $\delta = 0.05$ on 1000 realizations of the Bernoulli random variable with mean 0.9. **(L)**: We show the final value of $W$ depending on the choice of $m$ for the algorithms. The vanilla versions have exponential dependency on $m$, while the ✷ versions virtually always end up with $W \in \{0, \frac{1}{\delta}\}$. Additionally, we can confirm that the ✷ versions reject the null hypothesis for more values of $m$. **(R)**: Here we show the evolution of $W$ throughout the runs of the algorithms for $m = 0.86$. We can see that the Bernstein's testing algorithm already achieved the required wealth, but later lost it, unlike ✷-Bernstein's testing which stopped betting after reaching it. We can also see that towards the end, ✷-Hoeffding betting started betting very aggressively in order to have a chance to reach the desired wealth.

*Proof.* Consider Algorithm 1. We first show that $\ell^H$ is selected in a way that $\sum_{i=1}^n X_i$ required to reject the null hypothesis is smallest possible. That is:

$$\ell^H = \arg\min_{\lambda \in \mathbb{R}} \min_{X_1, \ldots, X_n \in [0,1]} \sum_{i=1}^n X_i, \quad \text{s.t.} \ \lambda \sum_{i=1}^n (X_i - m) - n\lambda^2/8 \geqslant \log \frac{1}{\delta} \ .$$

The constraint is $\sum_{i=1}^n X_i \geqslant nm + \log \frac{1}{\delta}/\lambda + n\lambda/8$. Minimizing the RHS over $\lambda$, the solution is $\lambda = \sqrt{8 \log \frac{1}{\delta}/n}$.

Now, consider Algorithm 2. By the same argument, $\ell_{\star}^H$ at time $t$ is minimizing $\sum_{i=t}^n X_i$ under the constraint that the null hypothesis is rejected. Consequently, the required $\sum_{i=1}^n X_i$ for Algorithm 2 is initially the same as for Algorithm 1, but it decreases whenever $\ell_{\star}^H$ changes with respect to the previous iteration. $\qquad\square$

**Corollary 5.** *Consider generating confidence intervals using Theorem 2 and a betting algorithm. The confidence intervals given by the betting Algorithm 2 are not-wider than the confidence intervals by Algorithm 1, all other things being equal. Moreover, there are cases when Algorithm 2 produces strictly smaller confidence intervals.*

*Remark* 6. Many popular confidence intervals including Hoeffding's, Bernstein's, and Bennet's can benefit from ✷-betting. A standard way to derive concentration inequalities is to bound the cumulant generating function. We have seen this in Hoeffding's inequality with $\mathbb{E}\left[\exp(\ell \cdot (X - \mu))\right] \leqslant \exp(\ell^2/8)$. In general, we get $\mathbb{E}\left[\exp(\ell \cdot (X - \mu))\right] \leqslant f(\ell)$. Then, $\ell$ is selected to minimize $\sum_{i=1}^n X_i$ under the condition that $\ell \sum_{i=1}^n (X_i - m) - \log f(\ell) \geqslant \log \frac{1}{\delta}$, which is precisely the reason why ✷-betting outperforms standard betting in Proposition 4. We demonstrate this in Figure 1.

We observe that ✷-algorithms usually end with either $0$ or $\frac{1}{\delta}$ wealth. This is a natural consequence of the adaptiveness, as we described in Figure 1 from both sides - if ✷ algorithm reaches the target, it stops betting. On the other hand, if the target is not reached yet, the bets become more and more aggressive, often resulting in bankruptcy. This is not a bad property. If we want to design confidence intervals with exact coverage $1 - \delta$, the following proposition shows that it is actually necessary.

**Proposition 7.** *Consider the process $W_i = W_{i-1}(1 + \ell_i \cdot (X_i - \mu))$ with $W_0 = 1$ and an algorithm for selecting bets $\ell_i$. If the algorithm always finishes with $W_n \in \{0, \frac{1}{\delta}\}$, then the algorithm falsely rejects the hypothesis $H_0(\mu) : \mu = \mu$ at confidence level $1 - \delta$ with probability precisely $1 - \delta$.*

*Proof.* By construction, $\mathbb{E}[W_n] = 1$. Let the algorithm halt with $W_n = \frac{1}{\delta}$ with probability $p$, then $\mathbb{E}[W_n] = \frac{p}{\delta}$, yielding $p = \delta$ as required. $\qquad\square$

## 5 STaR-Bets Algorithm

In the previous section, we have described the key concept of ✱-betting that can be implemented within the majority of common (finite time) betting schemes. It remains to choose a good betting scheme. So, here will introduce our new algorithm: STaR-Bets. Now, we will present an informal reasoning to derive a first version of our betting algorithm. Then, we prove formally that it produces intervals with the optimal width.

Let $X_1, \ldots, X_n \in [0, 1]$ be an i.i.d. sample of a random variable $X$ for which we aim to find a one-sided interval for the mean parameter. In the following, we also assume that $n \gg \log \frac{1}{\delta}$, that is, our sample is large enough. This is not a restrictive assumption. If it is violated, then the optimal confidence interval has width $\approx 1$ for a general distribution on $[0, 1]$ and we cannot hope to have a short interval anyway.

We use the processes from (2) of the form $W_{i+1} = W_i(1 + \ell(X_i - m))$ to refute the hypothesis $m = \mathbb{E}[X]$. So, we construct a betting strategy that aims at achieving at least $\frac{1}{\delta}$ wealth after $n$ rounds. First, we approximate the logarithm of the final wealth using a second-order Taylor expansion:

$$\sum_{i=1}^{n} \log(1 + \ell_i \cdot (X_i - m)) \approx \sum_{i=1}^{n} \left( \ell_i \cdot (X_i - m) - \frac{\ell_i^2}{2} \cdot (X_i - m)^2 \right) . \tag{4}$$

For the moment, we will assume that this is a good approximation, but eventually, we will derive it formally.

Define $S \triangleq \sum_{i=1}^{n}(X_i - m)$ and $V \triangleq \sum_{i=1}^{n}(X_i - m)^2$. From the r.h.s. of (4), the constant bet $\ell^*$ maximizing the approximate wealth is

$$\ell^* \triangleq \arg\max_{\ell} \ \ell S - \frac{\ell^2}{2} V = \frac{S}{V} \Rightarrow \max_{\ell} \ \ell S - \frac{\ell^2}{2} V = \frac{(\ell^*)^2}{2} V . \tag{5}$$

A reasonable approach is to try to estimate $\ell^*$ over time, in order to achieve a wealth close to the one in (5). This is roughly the approach followed in previous work [see, e.g., 8]. Instead, here we follow the approach suggested in Waudby-Smith and Ramdas [23]: We are only interested in the case where the (approximate) log-wealth reaches $\log \frac{1}{\delta}$. In other words, the outcome of the betting game is binary: We either reach the desired log-wealth for the candidate of the mean, $m$, and we reject the null hypothesis $H_0(m) : \mathbb{E}[X] = m$, or we fail to do so. This means that we want

$$\frac{(\ell^*)^2}{2} V = \log \frac{1}{\delta} \Leftrightarrow |\ell^*| = \sqrt{\frac{2 \log \frac{1}{\delta}}{V}} .$$

Estimating $V$ online, Shekhar and Ramdas [21] proved that this strategy will give asymptotically optimal confidence intervals almost surely. However, we are aiming for a finite-time guarantee, which requires a completely different angle of attack.

First of all, it might not be immediately apparent why we expressed (5) as function of $V$ only, while we could also get an $S$ term. The reason is that this is an easier quantity to estimate: Estimating $\frac{V}{n} \approx \mathbb{E}[(X - m)^2] \in \Theta(1)$ is easier than estimating $\frac{S}{n} \approx \mathbb{E}[X - m] \in \Theta(\frac{1}{\sqrt{n}})$ in the terms of relative error. Relative error is relevant because if we overestimate $\ell^*$ by a factor of 2, then we would end up with 0 (approximated) log-wealth.

Also, observe that $\ell^\star \in \Theta(\sqrt{\log \frac{1}{\delta}/n})$ which justifies the Taylor approximation in (4). Indeed, we aim to approximate $\ell^\star$ and $|X_i - m| \leqslant 1$, so the error of the Taylor approximation is

$$n \cdot \mathcal{O}\left( \left( \sqrt{\frac{\log \frac{1}{\delta}}{n}} \right)^3 \right) = \mathcal{O}\left( \log \frac{1}{\delta} \sqrt{\frac{\log \frac{1}{\delta}}{n}} \right) .$$

This error is of an order smaller than $\log \frac{1}{\delta}$ as long as $n \gg \log \frac{1}{\delta}$, as we have assumed. Recalling our goal of achieving a log-wealth of $\log \frac{1}{\delta}$, this means that the Taylor approximation is sufficiently good.

Let us now examine the approximate length of the confidence interval consisting of candidate means $m$ for which we did not reach log-wealth $\log \frac{1}{\delta}$:

$$\ell^* S - \frac{\ell^{*2}}{2} V \leqslant \log \frac{1}{\delta} \implies S \sqrt{\frac{2 \log \frac{1}{\delta}}{V}} \leqslant 2 \log \frac{1}{\delta} \implies S \leqslant \sqrt{2V \log \frac{1}{\delta}} . \tag{6}$$

| **Algorithm 3** Testing with Bets | **Algorithm 4** Testing with ✶-Bets |
|---|---|
| **Require:** i.i.d. $X_1, \ldots, X_n \in [0,1]$ | **Require:** i.i.d. $X_1, \ldots, X_n \in [0,1]$ |
| **Require:** $\alpha, \delta > 0$, $m \in [0,1]$, $n$ | **Require:** $\alpha, \delta > 0$, $m \in [0,1]$, $n$ |
| $\quad V \leftarrow 0, \text{lgW} \leftarrow 0$ | $\quad V \leftarrow 0, \text{lgW} \leftarrow 0$ |
| $\quad$ **for** $1 \leqslant t \leqslant n$ **do** | $\quad$ **for** $1 \leqslant t \leqslant n$ **do** |
| $\qquad v \leftarrow \left( V/(t-1) + \frac{10 \log \frac{8}{\alpha} n}{(t-1)^2} \right) \wedge 1$ | $\qquad v \leftarrow \left( V/(t-1) + \frac{10 \log \frac{8}{\alpha} n}{(t-1)^2} \right) \wedge 1$ |
| $\qquad \ell \leftarrow \sqrt{2 \log \frac{1}{\delta}/(nv)} \wedge 1$ | $\qquad \ell \leftarrow \sqrt{2(\log \frac{1}{\delta} {\color{red}-\text{lgW}})/((n{-}t+1)v)} \wedge 1$ |
| $\qquad \text{lgW} \leftarrow \text{lgW} + \log(1 + \ell(X_i - m))$ | $\qquad \text{lgW} \leftarrow \text{lgW} + \log(1 + \ell(X_i - m))$ |
| $\qquad V \leftarrow V + (X_i - m)^2$ | $\qquad V \leftarrow V + (X_i - m)^2$ |
| $\qquad$ **if** $\text{lgW} \geqslant \log \frac{1}{\delta}$ **then** Rejected | $\qquad$ **if** $\text{lgW} \geqslant \log \frac{1}{\delta}$ **then** Rejected |

We can further approximate $V \approx \mathbb{E}[V] = n\mathbb{V}[X] + \mathbb{E}[S]^2/n \approx n\mathbb{V}[X]$, since $n\mathbb{V}[X] \in \Theta(n)$, while $\mathbb{E}[S]^2/n \in \Theta(1)$, then the bound from (6) matches the width from the Bernstein's inequality, which is the one of a Normal approximation and thus cannot be improved.

The only missing ingredient is how to estimate $V$ over time. Recalling that we aim for a low relative error, after seeing $t$ outcomes we will use the estimator $V/n \approx \frac{\sum_{i=1}^t (X_i - m)^2}{t} + \frac{n \log \frac{1}{\alpha}}{t^2}$, where the second term is carefully constructed to guarantee that the estimate has a small relative error with high probability (depending on $\alpha$) uniformly over $m$ and $1 \leqslant t \leqslant n$.

The final algorithm is shown in Algorithm 3, where we run the above betting procedure for a possible values of $m \in [0,1]$ and reject them as mean candidates if the log-wealth is at least $\log \frac{1}{\delta}$.

**Theorem 8.** *For every random variable $X \in [0,1]$ with mean $\mu$ and variance $\sigma^2 > 0$, every $\alpha, \delta \in (0,1)$, and $c > 0$, there is $n_0$ depending on $\alpha, \delta, \sigma^2, c$, such that for all $n \geqslant n_0$, Algorithm 3 rejects at confidence level $1 - \delta$ every $m$ satisfying*

$$m \leqslant \frac{\sum_{i=1}^n X_i}{n} - \sigma \sqrt{\frac{(2+c) \log \frac{1}{\delta}}{n}} \ . \tag{7}$$

*with probability at least $1 - \alpha$. Furthermore, we have that $n_0 \in \mathcal{O}(c^{-4})$ when treating the other variables as constants.*

The proof is in Appendix A.

**Corollary 9.** *Algorithm 3 can be used in the framework of Theorem 2 (using the discretization technique for constructing intervals) to construct confidence intervals of the width up to a $1 + o(1)$ factor diminishing with $n$.*

Our proposed algorithm is ✶-Bets in Algorithm 4, the ✶-version of Algorithm 3. We discuss the implementation details in Appendix D.

We remark that while the algorithm is tailored to confidence intervals, it naturally produces a confidence sequence as a by-product. More concretely, we do not need to construct the test process $W_1, \ldots W_n$ from Theorem 2 at once and then collect the non-rejected mean candidates. We can construct the test processes sequentially when new samples arrive and after each of which we can recompute the confidence interval. A mild modification (either replace Markov's inequality by Ville's inequality, or do some brain gymnastics) of the original argument shows that all the confidence intervals contain the mean simultaneously with probability $1 - \delta$ in the frequentist sense. Straightforward modification of Theorem 8 shows that the width of the confidence interval after observing $t$ samples is (up to a constant factor) $\sqrt{\log \frac{1}{\delta} n/t^2}$. This being said, we believe that there are generally better options when one needs confidence sequences as discussed in Section 2, such as [13].

We conclude this section by emphasizing that while we have proven an optimality result of Algorithm 3 in Theorem 8, and other optimality results of ✶-technique in Proposition 4, when they are combined in Algorithm 4, we were not able to prove these optimality properties, and we only have empirical evidence of the superiority over the competitors. The validity of the hypothesis testing (and thus of the resulting confidence interval) still holds by Theorem 2.

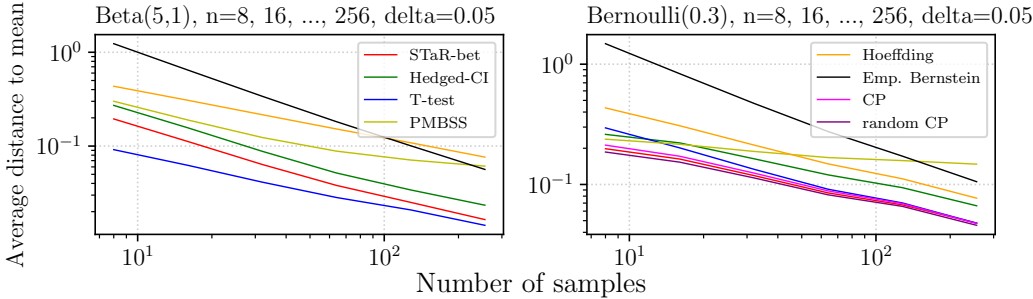

Figure 2: We directly compare the widths of the confidence intervals. Note the $\log-\log$ scale. For all the methods and every $n = 8, 16, \ldots, 256$, we have estimated the mean $1000\times$ of a fresh realization of the corresponding random variable and plotted the average distance to the mean. (L): When estimating the mean of beta distribution, we observe that that with increasing $n$, we are getting closer to the performance of T-test. (R): When estimating Bernoulli mean, the performance of �-Bets is very similar to the specialized optimal methods.

## 6 Experiments

Now we provide some experiments suggesting that ✻-Bets yields shorter confidence intervals than alternative methods. Here, we provide a "teaser" of our experiments, while the extensive experimental evaluation is in Appendix C. We identified several methods as our direct or indirect competitors and will briefly discuss them.

- Confidence interval derived from the T-test. This is a widely used confidence interval in practice, but it does not have guaranteed coverage in general. We show that our ✻-Bets algorithm is competitive with the T-test and has guaranteed coverage for $X \in [0,1]$.

- Clopper-Pearson [5] is the best deterministic confidence interval for a binomial sample (sum of Bernoulli random variables). Nevertheless, ✻-Bets often produces shorter intervals.

- Randomized Clopper-Pearson [22] is the optimal binomial confidence interval. ✻-Bets is still competitive.

- Hedged-CI [23] is a confidence interval based on betting and currently the best known algorithm for constructing confidence intervals. It is very similar to our Bets algorithm from 3 with comparable performance. The difference lies in the fact the we estimate $\mathbb{E}[(X-m)^2]$, while [23] estimates $\mathbb{E}[(X-\mu)^2]$. However, our ✻-Bets is significantly stronger.

- Hoeffding's inequality and the empirical Bernstein bound are the standard ways to construct confidence intervals, and so we include them in the experiments. Empirical Bernstein bound is usually weak because of the additive terms. Hoeffding's inequality is generally weak, but when $\mathbb{V}[X] \approx \frac{1}{2}$, it is competitive with some of the methods, but not with ✻-Bets.

- The method[4] of Phan et al. [16] was state of the art at the time of introduction, aiming at short intervals for very small $(10-50)$ sample sizes. We show that ✻-Bets is is stronger even in this regime. Furthermore, this method does not scale well to large samples.

First, we perform several experiments with Beta and Bernoulli distribution to quickly assess the competing methods. We show in Figure 2 how the widths of the confidence intervals evolve as we increase $n$ and conclude, that from our direct competitors, Hedged-CI of Waudby-Smith and Ramdas [23] is the strongest one, thus we use it in our further experiments. In Figure 3, we provide teaser for the extensive experimental evaluation in Appendix C. Specially, we introduce a CDF figure that shows the distribution of the lower-confidence bounds over 1000 repetitions of the experiments with the same setting, which allows for a systematic comparison of the methods.

---

[4]It is called practical mean bounds for small samples, so we abbreviate it as PMBSS.

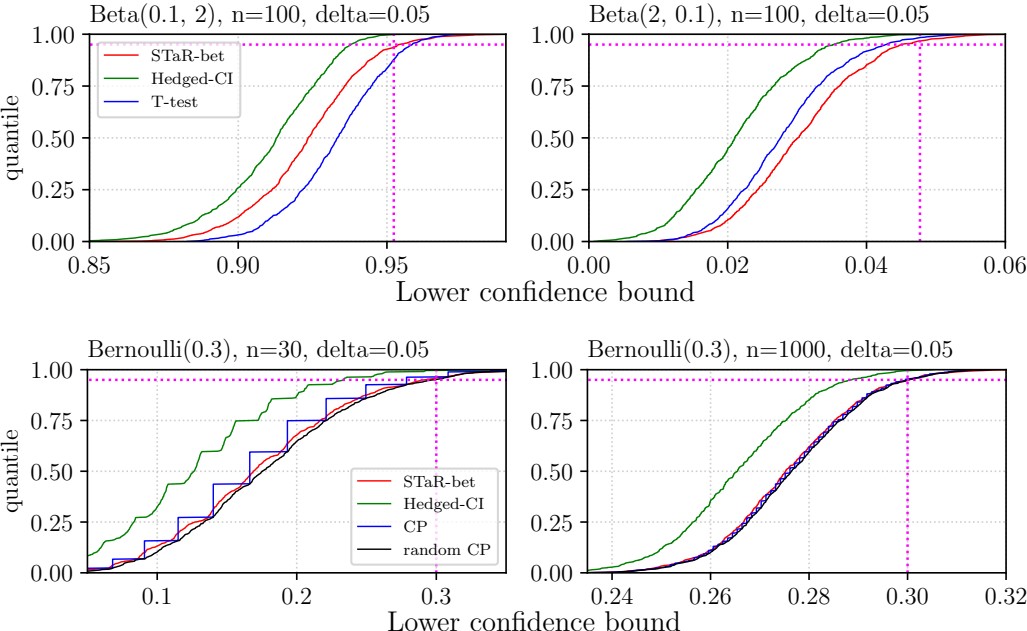

Figure 3: CDF figure: When a curve corresponding to a method passes through a point $(x, y)$, it means that the $y$-fraction (of 1000 repetitions) of lower confidence bounds was smaller than $x$. The vertical magenta line shows the mean position, and the vertical one shows the $1 - \delta$ quantile. We can see that in both cases, ✸-Bets passes through the intersection, implying that the coverage is $\approx 1 - \delta$. (Top) Estimation of the mean of Beta distribution (L): we can see that T-test produces shorter intervals than ✸-Bets, but that it has significantly smaller coverage than claimed. (R): Here, ✸-Bets produces shortest intervals. (Bottom) Estimation of the mean of a Bernoulli distribution using 30 (L) and 1000 (R) samples. We observe that in the low sample regime, ✸-Bets is mildly worse than the unbeatable randomized Clopper-Pearson, arguably better than standard Clopper-Pearson, and significantly better than Hedging of [23]. In the regime of larger samples, we can see that ✸-Bets stays very close to the optimal intervals, while the competitor is still significantly worse.

**Conclusions:** We have introduced ✸-technique in the construction of confidence intervals that directly (strictly) improves many betting algorithms and concentration inequalities, such as the ones of Hoeffding and Bernstein. Then, we have proposed a new betting algorithm for which he have proven that it can construct confidence intervals of the optimal length up to diminishing factor. While the ✸-technique is powerful in experiments, we have only proven that it never hurts (certain class of algorithms) and that it usually helps. How much it helps remains an open theoretical question.

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

## A   Proof of Theorem 8

Before we prove the theorem, we introduce several propositions we use in the sequel. We start with a (time uniform) version of Bennett's inequality.

**Proposition 10** (part of Lemma 1 of [2]). *Let $X_1, \ldots, X_n \leqslant b$ be i.i.d. real-valued random variables and let $b' = b - \mathbb{E}[X]$. For any $\delta \in (0, 1)$, simultaneously for all $1 \leqslant t \leqslant n$, we have*

$$\sum_{i=1}^{t}(X_i - \mathbb{E}[X]) \leqslant \sqrt{2n\mathbb{V}[X]\log\frac{1}{\delta}} + \frac{b'}{3}\log\frac{1}{\delta}\,.$$

*More generally, let $X_1, \ldots X_n < b$ be a sequence of martingale differences (i.e., $\mathbb{E}[X_i \mid X_1, \ldots, X_{i-1}] = 0$ for all $i$). Let $nV = \sum_{i=1}^{n}\mathbb{V}[X_i \mid X_1, \ldots, X_{i-1}]$. For any $\delta \in (0, 1)$, simultaneously for all $1 \leqslant t \leqslant n$, we have*

$$\sum_{i=1}^{t}X_i \leqslant \sqrt{2nV\log\frac{1}{\delta}} + \frac{b}{3}\log\frac{1}{\delta}\,.$$

*Remark* 11. The first part of the proposition is implied by the second one and is the standard Freedman's inequality up to a factor of 2 in the last term. This factor was removed in [2], but the result was stated for an i.i.d. sequence (the first part of the lemma) and not for martingale differences. However, the proof of [2] applies also to the second part.

Now, we use it to bound the deviation of our second moment estimator from the mean for all $m$ and $t$ simultaneously with high probability.

**Proposition 12.** *Let $X_1, \ldots, X_n \in [0, 1]$ be i.i.d. random variables with variance $\mathbb{V}[x] = \sigma^2$. For any $\alpha \in (0, 1)$, simultaneously for all $1 \leqslant t \leqslant n$ and all $m \in [0, 1]$ we have all the following inequalities with probability at least $1 - \alpha$:*

$$\frac{\sum_{i=1}^{t}(X_i - m)^2}{t} + \frac{10n\log\frac{4}{\alpha}}{t^2} \leqslant \mathbb{E}[(X - m)^2]\left(1 + \sqrt{\frac{18n\log\frac{4}{\alpha}}{t^2\mathbb{E}[(X-m)^2]}} + \frac{11n\log\frac{4}{\alpha}}{t^2\mathbb{E}[(X-m)^2]}\right),$$

$$\frac{\sum_{i=1}^{t}(X_i - m)^2}{t} + \frac{10n\log\frac{4}{\alpha}}{t^2} \geqslant \mathbb{E}[(X - m)^2]\overbrace{\left(1 - \sqrt{\frac{18n\log\frac{4}{\alpha}}{t^2\mathbb{E}[(X-m)^2]}} + \frac{9n\log\frac{4}{\alpha}}{t^2\mathbb{E}[(X-m)^2]}\right)}^{\geqslant 0.5},$$

$$\left|\frac{\sum_{i=1}^{t}X_i^2}{t} - \mathbb{E}[X^2]\right| \leqslant \sqrt{\frac{2n\sigma^2\log\frac{4}{\alpha}}{t^2}} + \frac{\log\frac{4}{\alpha}}{3t},$$

$$\left|\frac{\sum_{i=1}^{t}X_i}{t} - \mathbb{E}[X]\right| \leqslant \sqrt{\frac{2n\sigma^2\log\frac{4}{\alpha}}{t^2}} + \frac{\log\frac{4}{\alpha}}{3t}\,. \tag{8}$$

*Proof.* We use Proposition 10 on random variables $X, -X, X^2, -X^2$ and union bound, using the fact that $\mathbb{V}[X^2] \leqslant \mathbb{V}[X]$ as $X \in [0, 1]$. This yields the latter two identities. Then, as $X^2 - 2mX + m^2 = (X - m)^2$, we have accumulated $1 + 2m \leqslant 3$ of the identical error terms, yielding

$$\left|\frac{\sum_{i=1}^{t}(X_i - m)^2}{t} - \mathbb{E}[(X - m)^2]\right| \leqslant \sqrt{\frac{18n\sigma^2\log\frac{4}{\alpha}}{t^2}} + \frac{\log\frac{4}{\alpha}}{t},$$

from which we express the stated bounds, using $\mathbb{E}[(X - m)^2] \geqslant \sigma^2$ and $\log\frac{4}{\alpha}/t \leqslant n\log\frac{4}{\alpha}/t^2$. The lower bound then follows from completing the square. □

Further, we use the uniform version of Bennett's inequality to bound the sum of the observations in the first $t$ rounds; again, for all $m$ simultaneously.

**Proposition 13.** *Let $X_1, \ldots, X_n \in [0, 1]$ be i.i.d. random variables with mean $\mu = \mathbb{E}[X]$ and variance $\sigma^2 = \mathbb{V}[X]$, and let $0 \leqslant \lambda_i \leqslant C$ be random variables such that $\lambda_i$ is independent of $X_j$ for $j \geqslant i$ for some positive constant $C$. With probability at least $1 - \alpha$ we have for all $m \in [0, \mu]$ and all $1 \leqslant t \leqslant n$ simultaneously that*

$$\sum_{i=1}^{t} \lambda_i(X_i - m) \leqslant C \left( \sqrt{2t\sigma^2 \log \frac{2}{\alpha}} + \frac{1}{3} \log \frac{2}{\alpha} + t(\mu - m) \right),$$

$$\sum_{i=1}^{t} \lambda_i(X_i - m) \geqslant -C \left( \sqrt{2t\sigma^2 \log \frac{2}{\alpha}} + \frac{1}{3} \log \frac{2}{\alpha} \right),$$

$$\left| \sum_{i=1}^{t} (X_i - \mu) \right| \leqslant C\sqrt{2t\sigma^2 \log \frac{2}{\alpha}} + \frac{C}{3} \log \frac{2}{\alpha} \, .$$

*Proof.* We decompose the random variables $\lambda_i(X_i - m)$ into a martingale difference sequence $M_i = \lambda_i(X - \mu)$ and a drift term $D_i = \lambda_i(\mu - m)$. We bound deterministically the drift term:

$$0 \leqslant \sum_{i=1}^{t} D_i \leqslant tC(\mu - m),$$

and so it holds for all $m$ and $t$ simultaneously. We bound the martingale using the Freedman's-style inequality from Proposition 10:

$$\mathbb{P} \left\{ \left| \sum_{i=1}^{t} M_i \right| \geqslant C\sqrt{2t\sigma^2 \log \frac{2}{\alpha}} + \frac{C}{3} \log \frac{2}{\alpha} \right\} \leqslant \alpha \, .$$

Adding up the two bounds concludes the proof. $\qquad \square$

Finally, we introduce a quadratic lower-bound of a logarithm near zero.

**Lemma 14.** *Let $c \geqslant \frac{1}{2}$ and $|x| \leqslant 1 - \frac{1}{2c}$, then*

$$\log(1 + x) \geqslant x - cx^2 \, .$$

*Proof.* We inspect the behavior of $f(x) = \log(1 + x) - x + cx^2$. First, $f(0) = 0$. Now we look at the derivatives $f'(x) = 1/(x + 1) - 1 + 2cx$ and $f''(x) = -1/(x + 1)^2 + 2c$. Setting $f'$ to zero, we get $x(1 - 2c - 2cx) = 0$, so the roots are $x_1 = 0$ and $x_2 = 1/(2c) - 1$, these points are local extremes, concretely a local minimum and local maximum respectively by the second derivative test, and so the inequality holds for $x \geqslant 1/(2c) - 1$. $\qquad \square$

Now we are ready to restate the main theorem we are to prove.

**Theorem 15** (Theorem 8 restated)**.** *For every random variable $X \in [0, 1]$ with mean $\mu$ and variance $\sigma^2 > 0$, every $\alpha, \delta \in (0, 1)$, and $c > 0$, there is $n_0$ depending on $\alpha, \delta, \sigma^2, c$, such that for all $n \geqslant n_0$, Algorithm 3 rejects at confidence level $1 - \delta$ every $m$ satisfying*

$$m \leqslant \frac{\sum_{i=1}^{n} X_i}{n} - \sigma\sqrt{\frac{(2 + c) \log \frac{1}{\delta}}{n}} \, . \tag{9}$$

*with probability at least $1 - \alpha$.*

*Proof.* Statement (9) holds true if the events from Proposition 12 and Proposition 13 (applied on $X_1, \ldots, X_{c_2 n}$, to be specified later) are met. Namely, Proposition 12 ensures that our estimator of $\mathbb{E}[(X - m)^2]$ is never too small and is consistent. It also ensures that empirical mean of $X$ converges to $\mu$. Proposition 13 provides a bound on the wealth in an early stage. We note that both of the propositions hold uniformly for all $m \in [0, \mu]$. Thus, we instantiate both of the bounds (with failure probabilities $\alpha/2$) and further assume that the events are met. Throughout the sketch, we will be

introducing new constants on the fly. For all constants, we have that we can choose a large enough $n_0$ to make them arbitrarily close to 0.

We fix $m \in [0, \mu]$ whose exact expression will be decided at the end of the proof. Let $Y = X - m$ and $\varepsilon = \mu - m$. Let $\ell_{\text{opt}} = \sqrt{\frac{2 \log \frac{1}{\delta}}{n(\sigma^2 + \varepsilon^2)}}$. By Lemma 14, for any constant $1/2 \geq c_1 > 0$, we have that $\log(1 + \ell Y) \geq \ell Y - (1/2 + c_1)\ell^2 Y^2$ for all $\ell \leq \sqrt{2}\ell_{\text{opt}}$.

**Analysis of the run of algorithm:** We split the $n$ steps into an arbitrarily (relatively) short "warm up" phase of $c_2 n$ steps, where things can go poorly, but the effect of this will be negligible. Then, there will be a "convergent phase" of $(1 - c_2)n$ steps, in which $(1 - c_3)\ell_{\text{opt}} \leq \ell_i \leq (1 + c_3)\ell_{\text{opt}}$. We briefly comment on the constants. By Lemma 14 it holds that $c_1 \leq 3\ell_{\text{opt}} = \mathcal{O}(1/\sqrt{n})$ as long as $\ell_{\text{opt}} < 0.1$. $c_2$ is arbitrary, so we can set it to $\mathcal{O}(n^{-\frac{1}{4}})$. By Proposition 12, we have $c_3 \leq \frac{1}{c_2}\left(\sqrt{18 \log \frac{8}{\alpha}/(\sigma^2 n)} + 11 \log \frac{8}{\alpha}/(\sigma^2 n)\right)$, and so given the choice of $c_2$, we have $c_3 \in \mathcal{O}(n^{-1/4})$.

**Warm up phase:** In this phase, we have $0 \leq \ell_i \leq \sqrt{2}\ell_{\text{opt}}$ per the second part of Proposition 12 and the fact that $\mathbb{E}[(X - m)^2] = \sigma^2 + \varepsilon^2$.

First, deterministically upper bound the quadratic term:

$$\sum_{i=1}^{c_2 n} \left(c_1 + \frac{1}{2}\right) \ell_i^2 Y_i^2 \geq 2n \left(c_1 + \frac{1}{2}\right) c_2 \ell_{\text{opt}}^2 = \left(c_1 + \frac{1}{2}\right) c_2 \frac{4 \log \frac{1}{\delta}}{\sigma^2 + \varepsilon^2}.$$

Next, by Proposition 13 applied on $X_1, \ldots, X_{c_2 n}$ and $C = \sqrt{2}\ell_{\text{opt}}$, we have

$$\sum_{i=1}^{c_2 n} \ell_i(X_i - m) \geq -\sqrt{\frac{4 \log \frac{1}{\delta}}{n(\sigma^2 + \varepsilon^2)}} \left(\sqrt{2 c_2 n \sigma^2 \log \frac{4}{\alpha}} + \frac{1}{3} \log \frac{4}{\alpha}\right)$$

$$\geq -2\sqrt{2 c_2 \log \frac{1}{\delta} \log \frac{4}{\alpha}} + \sqrt{\frac{4 \log \frac{1}{\delta} \log^2 \frac{4}{\alpha}}{9n(\sigma^2 + \varepsilon^2)}},$$

where we see that both the quadratic and linear term go to zero as $n$ increases and $c_2$ decreases.

**Convergent phase:** By Proposition 12, we have $(1 - c_3)\ell_{\text{opt}} \leq \ell_i \leq (1 + c_3)\ell_{\text{opt}}$. From the definition of $\ell_i$ we have that $\ell_n^2 \sum_{i=1}^n Y_i^2 \leq 2 \log \frac{1}{\delta}$. Thus,

$$\sum_{i=c_2 n+1}^{n} \ell_i^2 Y_i^2 \leq \sum_{i=1}^{n} \ell_i^2 Y_i^2 \leq \left(\frac{1 + c_3}{1 - c_3}\right)^2 \ell_n^2 \sum_{i=1}^{n} Y_i^2 \leq \left(\frac{1 + c_3}{1 - c_3}\right)^2 2 \log \frac{1}{\delta}.$$

Now, we add up the lower bounds from the warm-up phase and from the quadratic term in the convergent case:

$$\overbrace{-2\sqrt{2 c_2 \log \frac{1}{\delta} \log \frac{4}{\alpha}} + \sqrt{\frac{4 \log \frac{1}{\delta} \log^2 \frac{4}{\alpha}}{9n(\sigma^2 + \varepsilon^2)}} - \left(c_1 + \frac{1}{2}\right) c_2 \frac{4 \log \frac{1}{\delta}}{\sigma^2 + \varepsilon^2}}^{\text{Warm-up phase}} \overbrace{- \left(c_1 + \frac{1}{2}\right) \left(\frac{1 + c_3}{1 - c_3}\right)^2 2 \log \frac{1}{\delta}}^{\text{Quadratic term in the convergent phase}}$$

$$= -(1 + c_4) \log \frac{1}{\delta},$$

for some $c_4 > 0$. Given the choices of constants above, we can have $c_4, c_5, c_6 \in \mathcal{O}(n^{-1/4})$ for the constants to come. Finally, we lower bound the log-wealth:

$$\sum_{i=1}^{n} \log(1 + \ell_i Y_i) \geq \ell_{\text{opt}}(1 - c_3)\left(\sum_{i=c_2 n+1}^{n} Y_i\right) - (1 + c_4) \log \frac{1}{\delta},$$

which is greater than $\log \frac{1}{\delta}$ if

$$\frac{\sum_{c_2 n+1}^{n} X_i}{(1 - c_2)n} - m \geq \overbrace{\frac{(2 + c_4)}{\sqrt{2}(1 - c_3)(1 - c_2)}}^{c_5 + \sqrt{2}} \sqrt{\frac{(\sigma^2 + \varepsilon^2) \log \frac{1}{\delta}}{n}}. \tag{10}$$

From Proposition 13, we have assumed the events

$$-\frac{1}{n}\sum_{i=1}^{c_2 n} X_i \geqslant -\frac{\sqrt{2c_2\sigma^2 \log\frac{4}{\alpha}}}{\sqrt{n}} - \mu c_2 - \frac{1}{3n}\log\frac{4}{\alpha} \geqslant -c_6/\sqrt{n} - \mu c_2$$

and

$$\frac{1}{n}\sum_{i=c_2 n+1}^{n}(X_i - \mu) \geqslant -\frac{\sqrt{2\sigma^2 \log\frac{4}{\alpha}}}{\sqrt{n}} - \frac{1}{3n}\log\frac{4}{\alpha} \geqslant -\frac{\sqrt{4\sigma^2 \log\frac{4}{\alpha}}}{\sqrt{n}} .$$

We now reveal our choice of $m$:

$$m = \frac{\sum_{i=1}^{n} X_i}{n} - (c_5 + \sqrt{2})(1 + \varepsilon/\sigma)\sqrt{\frac{\sigma^2 \log\frac{1}{\delta}}{n}} - \frac{c_6 + c_2\sqrt{4\sigma^2 \log\frac{4}{\alpha}}}{\sqrt{n}} .$$

We can now show the value of $m$ we have selected satisfies (10) and so it will result in a log-wealth bigger than $\frac{1}{\delta}$. Observe that

$$\frac{\sum_{i=c_2 n+1}^{n} X_i}{(1 - c_2)n} \geqslant (1 + c_2)\frac{\sum_{i=c_2 n+1}^{n} X_i}{n}$$

$$= \frac{\sum_{i=1}^{n} X_i}{n} + c_2\left(\frac{\sum_{i=c_2 n+1}^{n} X_i}{n} - \mu\right) + c_2\mu - \frac{\sum_{i=1}^{c_2 n} X_i}{n} .$$

Hence, we have

$$\frac{\sum_{i=c_2 n+1}^{n} X_i}{(1 - c_2)n} - m \geqslant (c_5 + \sqrt{2})\sqrt{\frac{(\sigma^2 + \varepsilon^2)\log\frac{1}{\delta}}{n}} .$$

Moreover, observe that these last inequalities apply for all $m' \leqslant m$.

Thus, for the chosen $m$, we have reached log-wealth $\log\frac{1}{\delta}$ and so it is rejected. Finally, we have assumed the event $\left|\frac{1}{n}\sum_{i=1}^{n} X_i - \mu\right| \lesssim 1/\sqrt{n}$ from (8). So, recalling that $\varepsilon = \mu - m$, for our choice of $m$ we also have $\varepsilon \approx 1/\sqrt{n}$. Hence, we can make $(c_5 + \sqrt{2})(1 + \varepsilon/\sigma)$ arbitrarily close to $\sqrt{2}$ and $c_2, c_6$ arbitrarily close to 0, finishing the proof of the main part. Given our tracking of constants, we can see that the final leading term is off by a factor $1 + \mathcal{O}(n^{-1/4})$, concluding the second statement. $\qquad\square$

## B  Bernstein betting

We derive the algorithm for Bernstein testing referenced in Figure 1 and show that ✶-ing it is a strict improvement. The steps are analogical to the Hoeffding's betting derivation. We note that the provided version of Bernstein's inequality is mildly weaker than the standard one in the interest of simplicity.

**Lemma 16** (Bernstein simplified). *Let $X \in [0, 1]$ with mean $\mu = \mathbb{E}[X]$ and variance $\sigma^2 = \mathbb{V}[X]$, then*

$$\mathbb{E}\left[e^{\ell \cdot (X - \mu) - \sigma^2\ell^2}\right] \leqslant 1, \quad \forall \ell \in \mathbb{R} . \tag{11}$$

*Furthermore, the sequence $W_0 = 1$, $W_{i+1} = W_i \exp\left(\ell \cdot (X_i - \mu) - \sigma^2\ell^2\right)$ for $i \geqslant 0$ is a test process.*

**Proposition 17.** *Let $m \in [0, 1]$. Whenever Algorithm 5 rejects the null hypothesis $H_0(m) : m = \mathbb{E}[X]$, then so does Algorithm 6 if they share the realizations $X_t$, $1 \leqslant t \leqslant n$.*

*Proof.* Consider Algorithm 5. We first show that $\ell^B$ is selected in a way that $\sum_{i=1}^{n} X_i$ required to reject the null hypothesis is smallest possible. That is:

$$\ell^H = \arg\min_{\lambda \in \mathbb{R}} \min_{X_1, \ldots, X_n \in [0, 1]} \sum_{i=1}^{n} X_i, \quad \text{s.t. } \lambda\sum_{i=1}^{n}(X_i - m) - n\sigma^2\lambda^2 \geqslant \log\frac{1}{\delta} .$$

The constraint is $\sum_{i=1}^{n} X_i \geqslant nm + \log \frac{1}{\delta}/\lambda + n\sigma^2 \lambda$. Minimizing the RHS over $\lambda$, the solution is $\lambda = \sqrt{\log \frac{1}{\delta}/(n\sigma^2)}$.

Now, consider Algorithm 6. By the same argument, $\ell^B_{\bigstar}$ at time $t$ is minimizing $\sum_{i=t}^{n} X_i$ under the constraint that the null hypothesis is rejected. Consequently, the required $\sum_{i=1}^{n} X_i$ for Algorithm 6 is initially the same as for Algorithm 5, but it decreases whenever $\ell^B_{\bigstar}$ changes with respect to the previous iteration. $\qquad\square$

---

**Algorithm 5** Bernstein testing

**Require:** i.i.d. $X_1, \ldots, X_n \in [0,1]$
**Require:** $\delta > 0$, $m \in [0,1]$, $n$, $\sigma^2 = \mathbb{V}[X]$
  $W \leftarrow 1$
  **for** $t = 1 \ldots n$ **do**
    $\ell^B \leftarrow \sqrt{\left(\log \frac{1}{\delta}\right)/(n\sigma^2)}$
    $W \leftarrow W \exp\left(\ell^B \cdot (X_t - m) - (\sigma\ell^B)^2\right)$
  **if** $W \geqslant \frac{1}{\delta}$ **then**
    Reject $H_0(m) : m = \mathbb{E}[X]$

---

**Algorithm 6** $\bigstar$- Bernstein testing

**Require:** i.i.d. $X_1, \ldots, X_n \in [0,1]$
**Require:** $\delta > 0$, $m \in [0,1]$, $n$, $\sigma^2 = \mathbb{V}[X]$
  $W \leftarrow 1$
  **for** $t = 1 \ldots n$ **do**
    $\ell^B_{\bigstar} \leftarrow \sqrt{\left(\log \frac{1}{W\delta}\right)_+/((n-t+1)\sigma^2)}$
    $W \leftarrow W \exp\left(\ell^B_{\bigstar} \cdot (X_t - m) - (\sigma\ell^B_{\bigstar})^2\right)$
  **if** $W \geqslant \frac{1}{\delta}$ **then**
    Reject $H_0(m) : m = \mathbb{E}[X]$

---

## C Experiments

We provide CDF plots as we believe they contain most information about the behavior of the confidence interval. We repeat the description from the main paper.

In short, the more the curve is to the right, the better it is.

Every algorithm provides a lower bound on the mean parameter of the corresponding distribution. We repeat the experiment 1000 times and for every algorithm plot the empirical CDF of the produced lower bounds. I.e., a curve passing through point $(x, y)$ should be understood as $y-$fractions of the lower bounds are smaller than $x$. We include a vertical and a horizontal magenta line representing the mean (vertical) and $1 - \delta$ (horizontal) as the desired coverage. The eCDF of the algorithm passes the vertical line at point $(\mu, \delta')$, where $\mu$ is the true mean and $1 - \delta'$ is the empirical coverage. We have zoomed in to a box centered at $(\mu, 1 - \delta)$ to see what is the coverage; also, we have added black vertical lines corresponding to $0.95-$one-sided confidence intervals. If the eCDF meets the vertical magenta line above (resp. below) the black line, it has coverage smaller (resp. bigger) than $1 - \delta$ at confidence level $0.95$. All algorithms apart from T-test have guaranteed coverage at least $1 - \delta$, so if they occur under the black line, it is by a chance.

### C.1 Experimental results

Here, we provide general experimental results. We always show STaR bet from Algorithm 4 with details from D. Hedged-CI is from [23] with the default settings. T-test and (randomized) Clopper-Pearson intervals are standard.

**Bernoulli:** Here, we can see that $\bigstar$ is performing very closely to randomized Clopper-Pearson and outperforming standard Clopper-Pearson on small sample sizes. On the larder sample sizes, the performance of all three algorithms become very similar, significantly outperforming Hedge-CI.

**Beta:** Here, we compete with T-test based confidence interval with no formal guarantees. We can see that as long as $a > b$ ($a, b$ are parameters of the beta distributions), T-test is outperforming $\bigstar$; however, it clearly has larger coverage than $1 - \delta$, violating the principles of confidence intervals. When $b > a$, $\bigstar$ usually provides shorter intervals than T-test. Hedged-CI provides much larger ones. With increasing $n$, the performance of T-test and of $\bigstar$ become essentially identical.

**Coverage:** In the majority of cases, the coverage of $\bigstar$ is statistically indistinguishable from $1 - \delta$. Coverage of Hedged-CI is usually $1$.

**Bernoulli, delta = 0.05, averaged over 1000 runs**

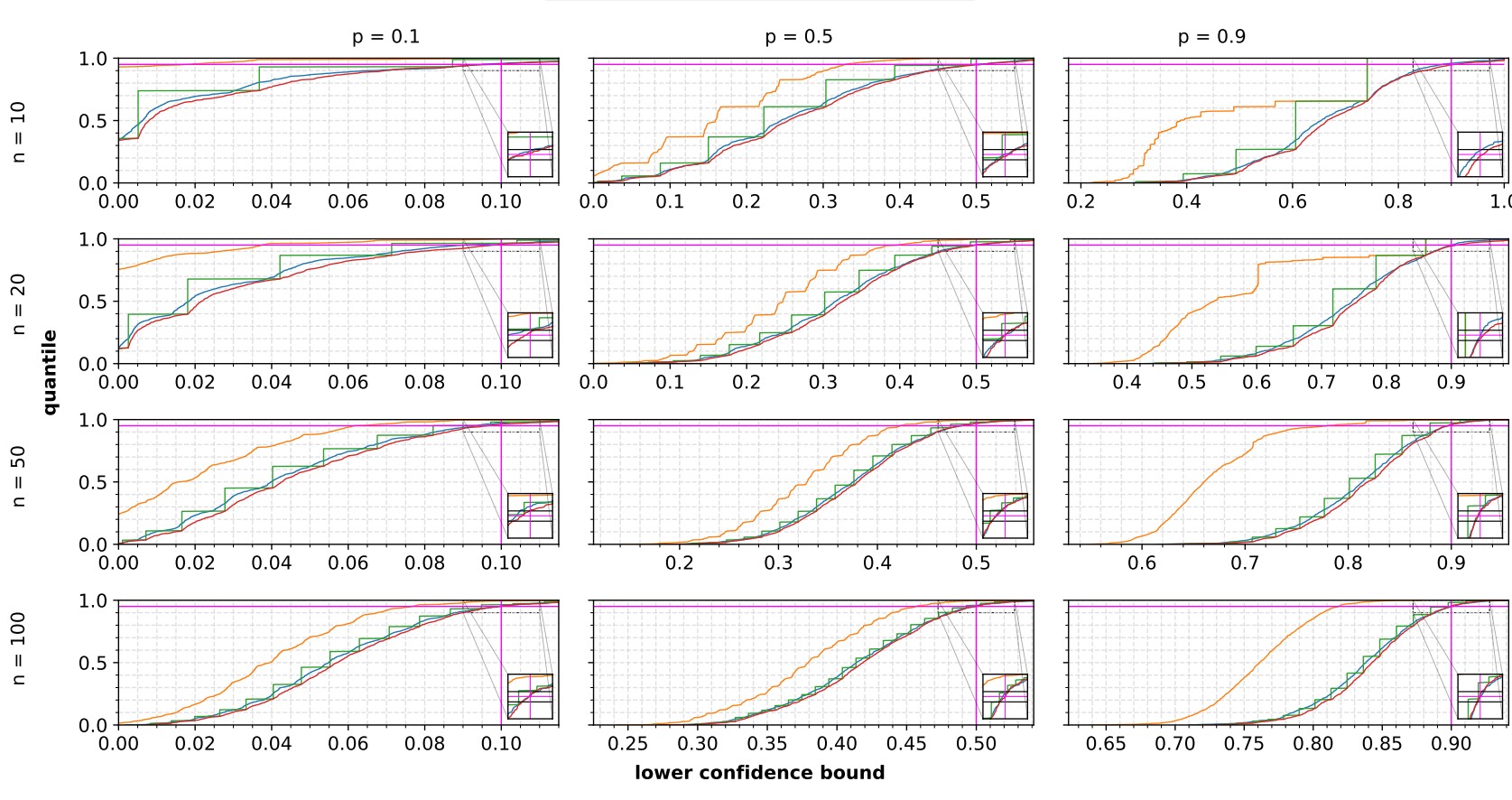

**Bernoulli, delta = 0.05, averaged over 1000 runs**

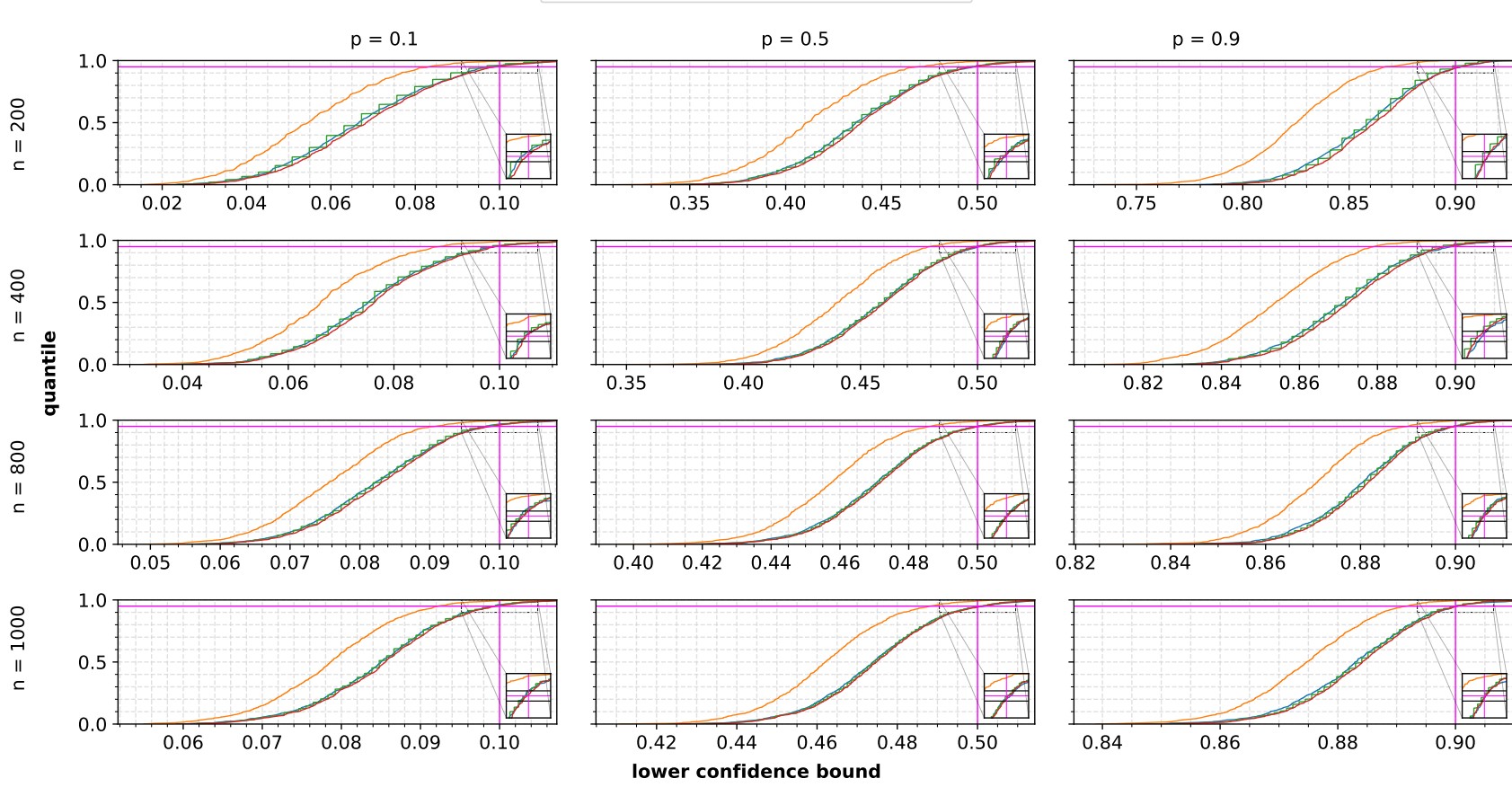

**Beta,  n = 10,  delta = 0.05, averaged over 1000 runs**

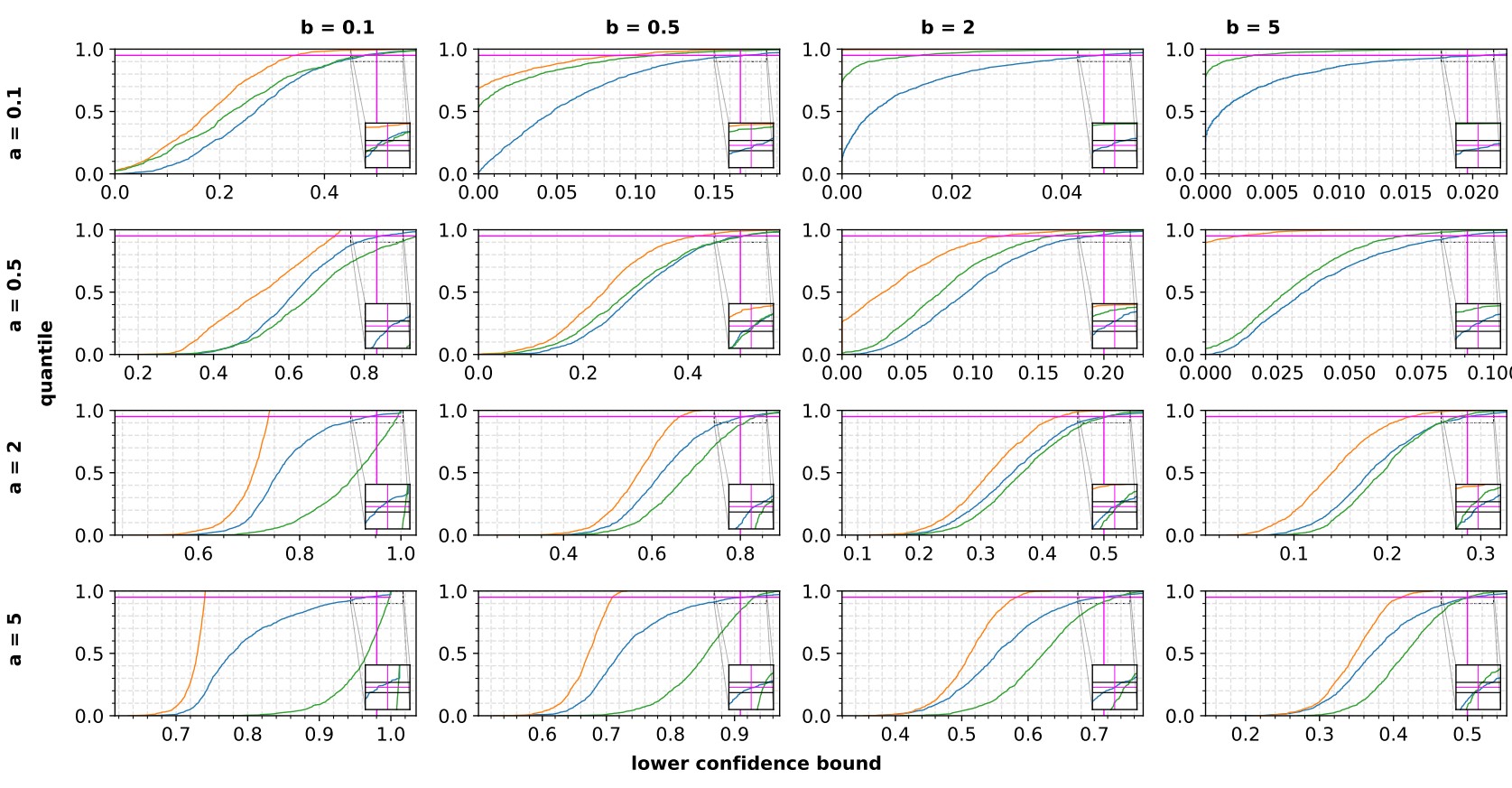

**Beta, n = 50, delta = 0.05, averaged over 1000 runs**

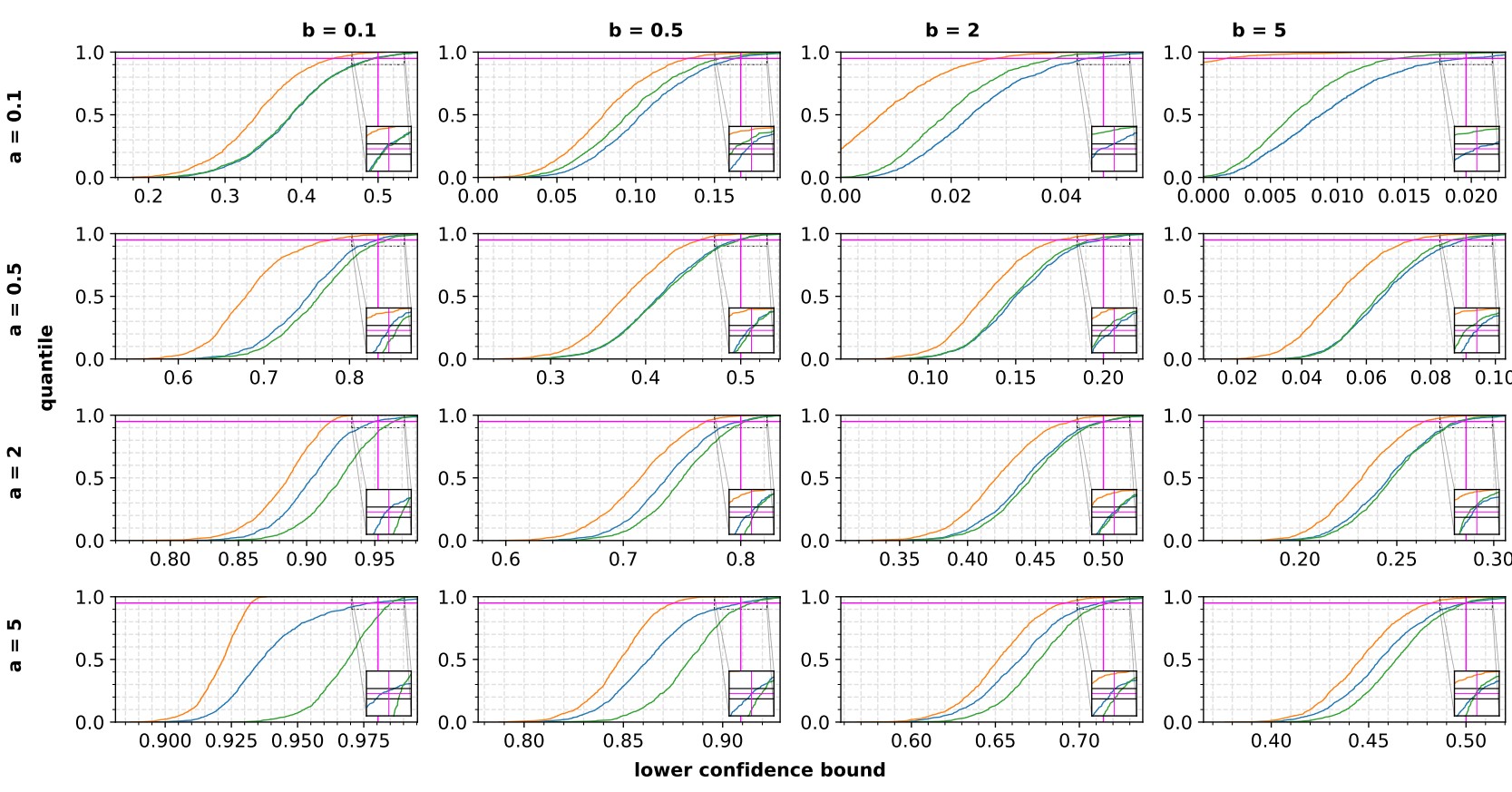

**Beta, n = 500, delta = 0.05, averaged over 1000 runs**

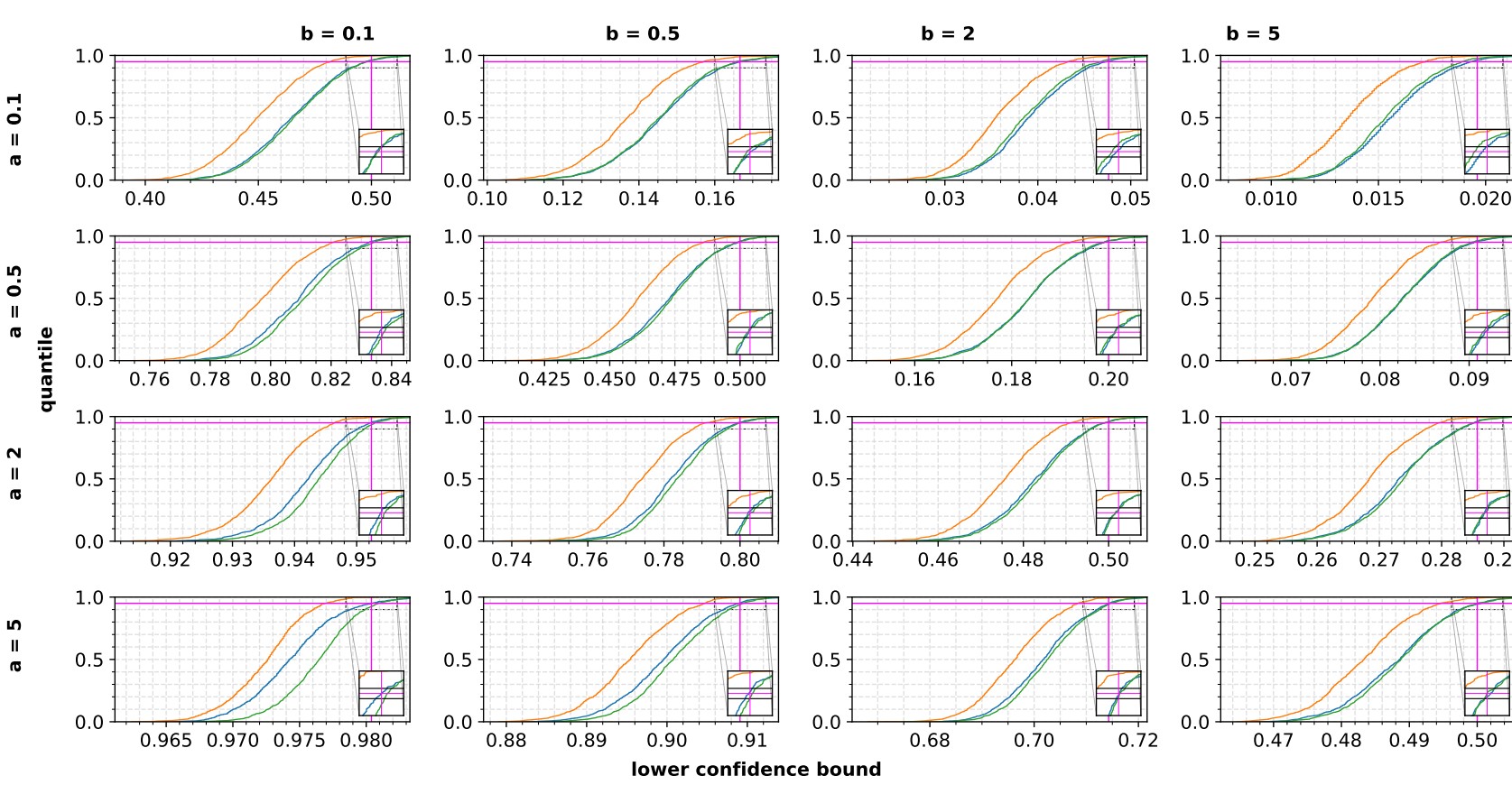

## C.2 Influence of $\delta$

The majority of the experiments are made with $\delta = 0.05$; here, we present the subset of the experiments with different values of $\delta$. The results follow the patterns observed with $\delta = 0.05$.

**Bernoulli, delta = 0.001, averaged over 1000 runs**

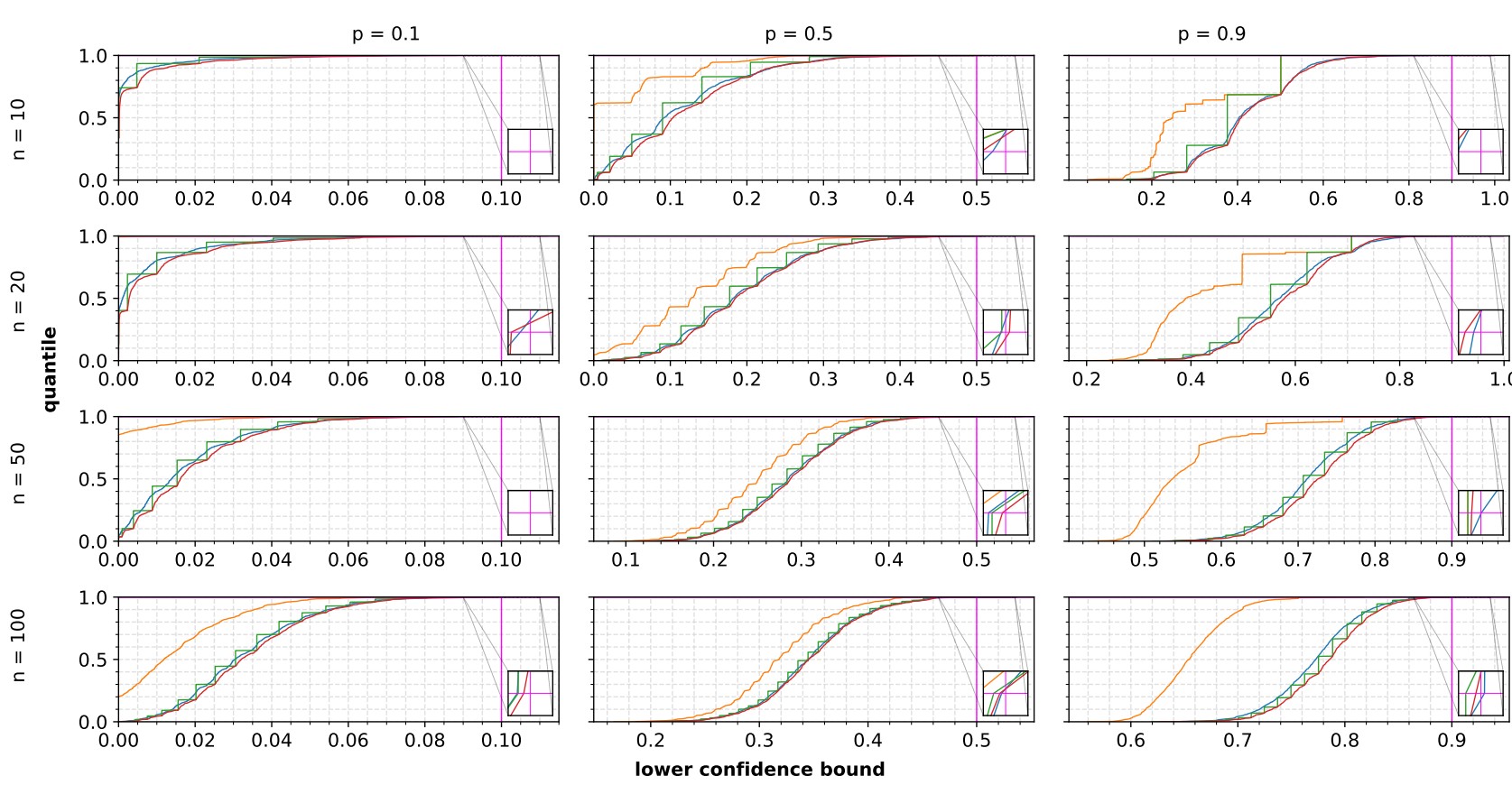

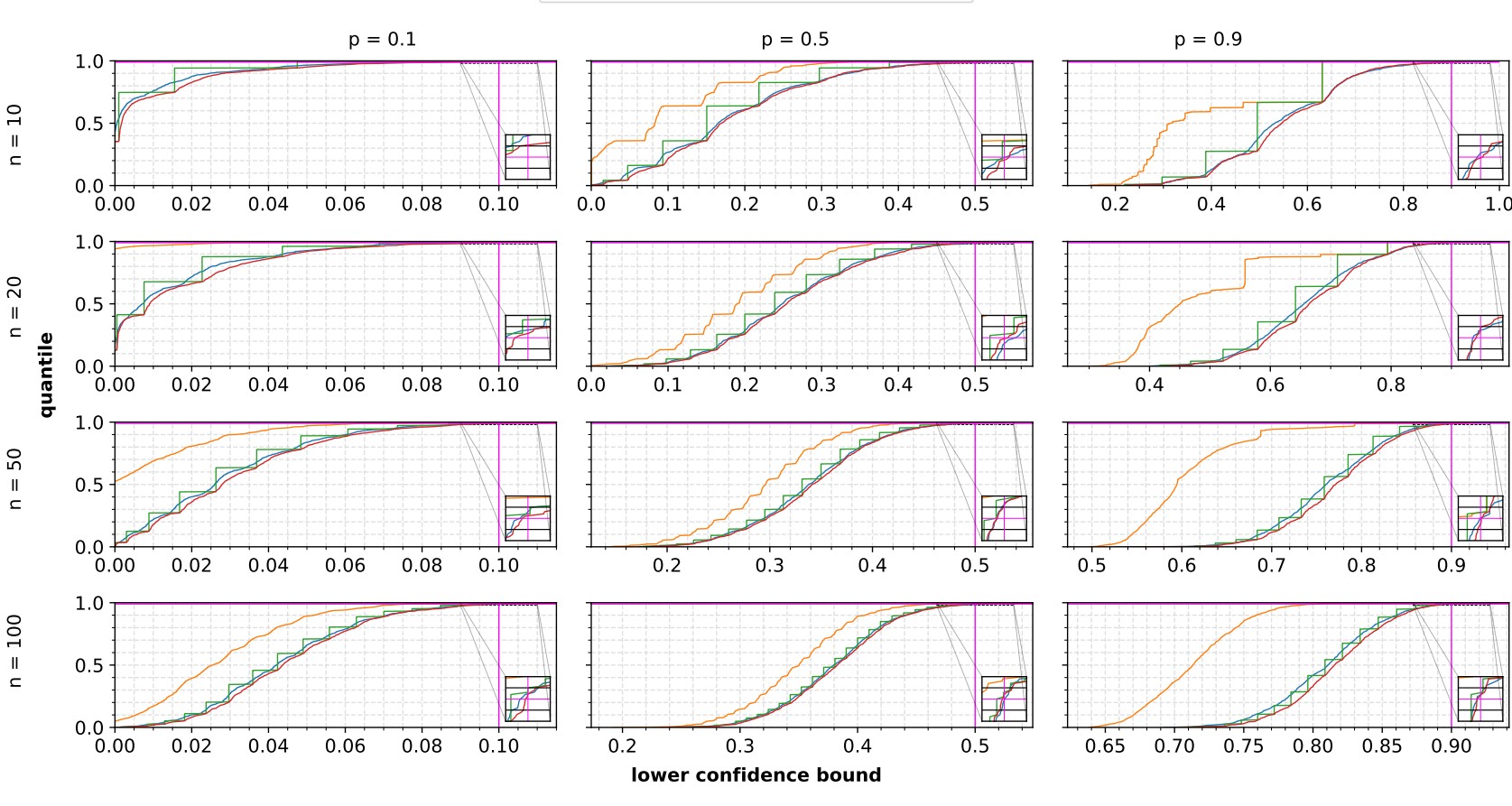

**Bernoulli, delta = 0.01, averaged over 1000 runs**

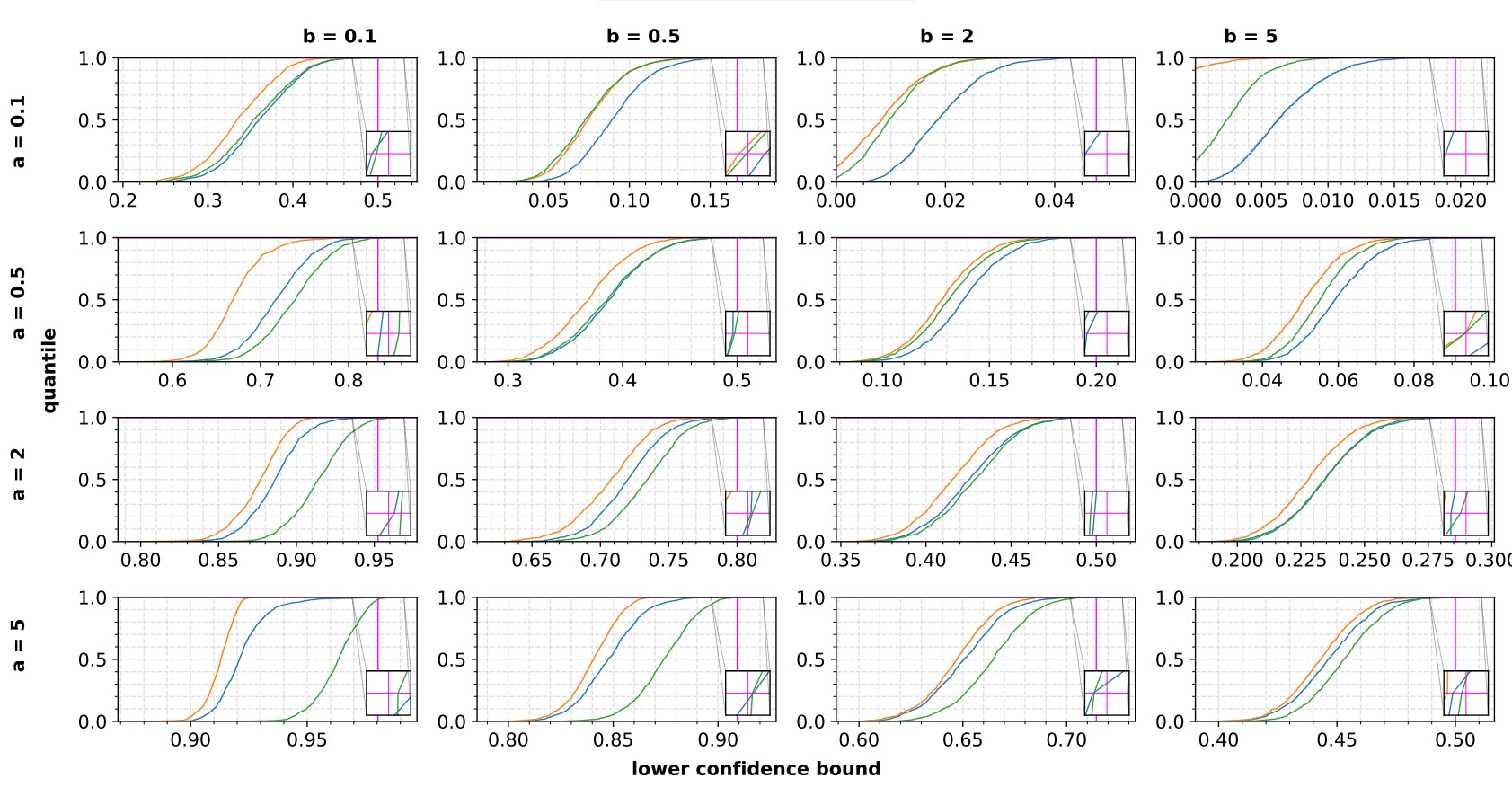

**Beta, n = 100, delta = 0.001, averaged over 1000 runs**

**Beta, n = 100, delta = 0.001, averaged over 1000 runs**

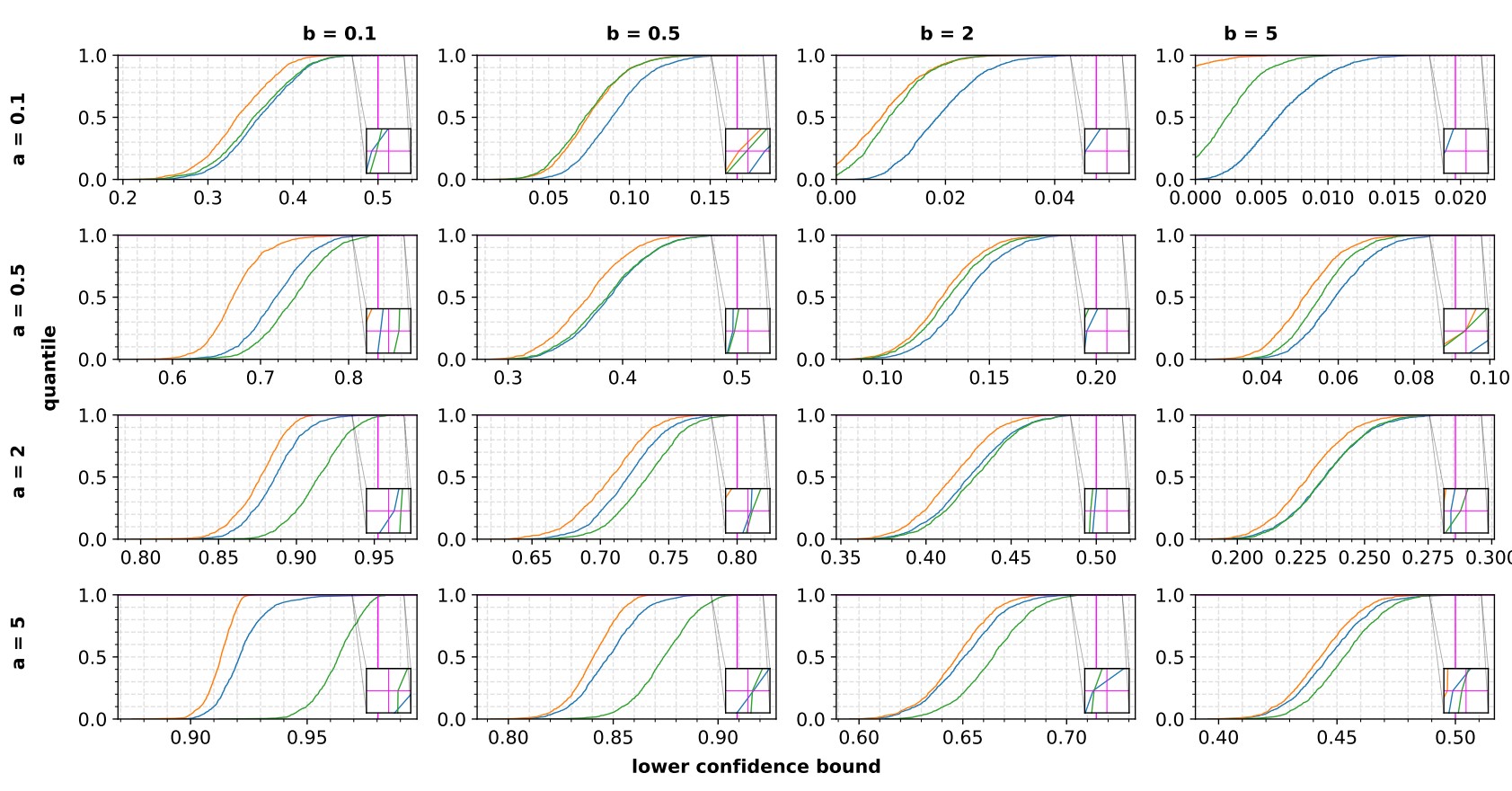

## C.3 Selection of c

As discussed in Appendix D.3, we estimate the expectation $\mathbb{E}[(X - m)^2]$ using the empirical mean with an additive $cmn/(t-1)^2$ term, where $t$ is the current round, $n$ is the number of rounds and $c$ is a free parameter. We provide experiments suggesting that the algorithm is not so sensitive about the choice, especially as $n$ increases, and we choose $c = 1$ as a natural choice to not overfit on our benchmark distributions.

In the following experiments, we used Algorithm 4 with the details in Appendix D sweeping over exponentially spaced grid of values of $c$.

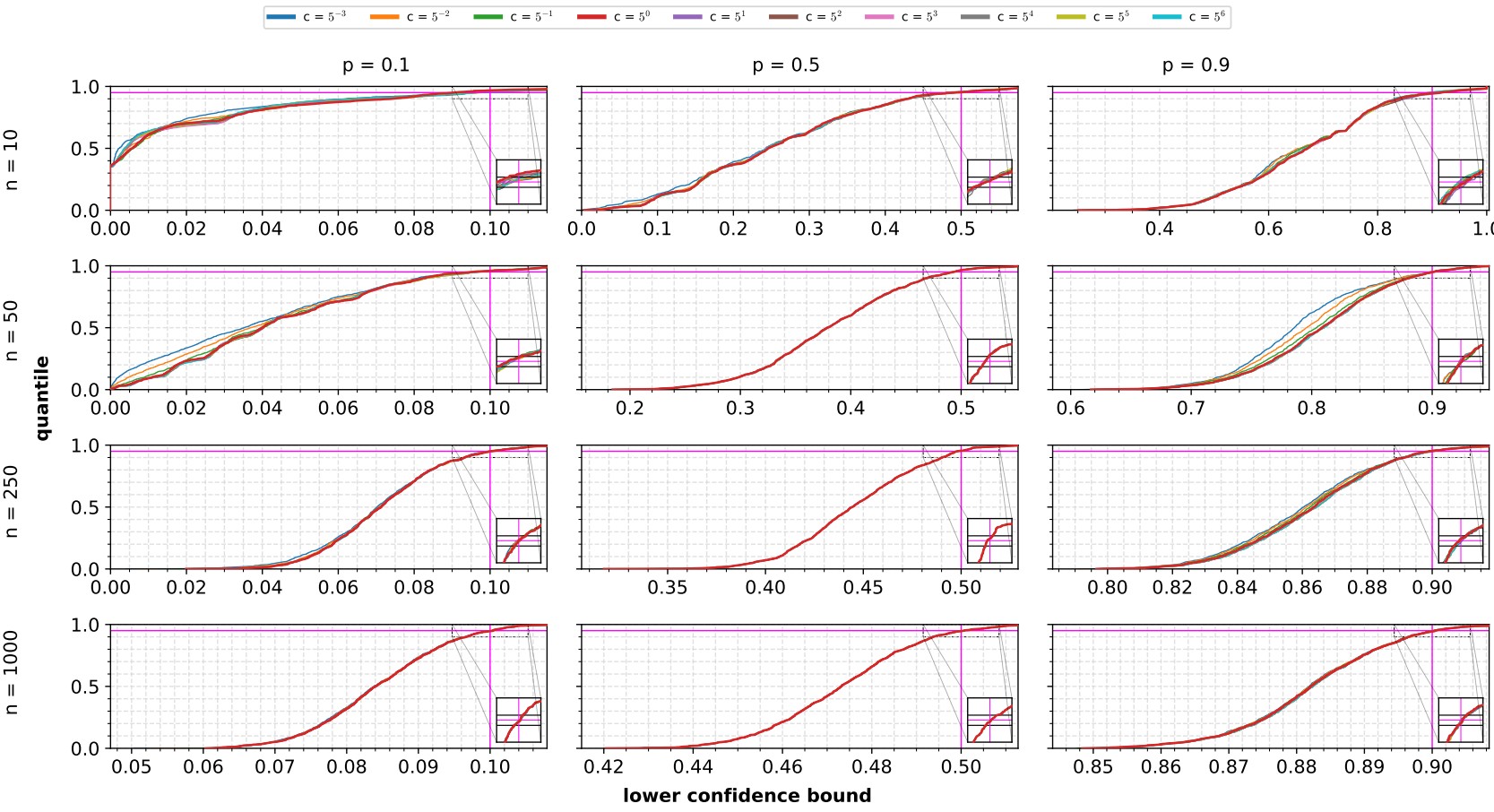

**Bernoulli, delta = 0.05, averaged over 1000 runs**

**Beta,  n = 10,  delta = 0.05, averaged over 1000 runs**

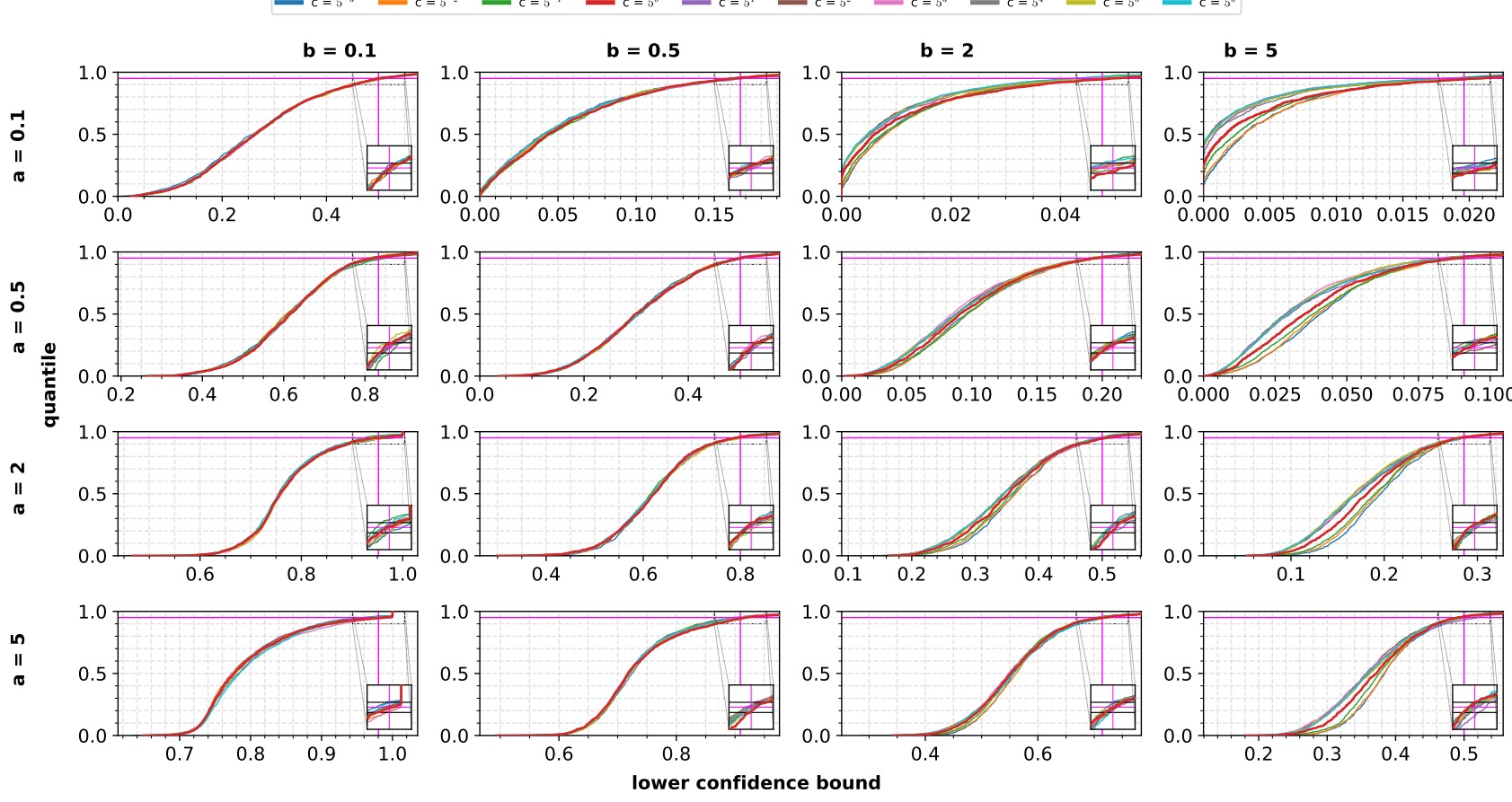

lower confidence bound

**Beta, n = 50, delta = 0.05, averaged over 1000 runs**

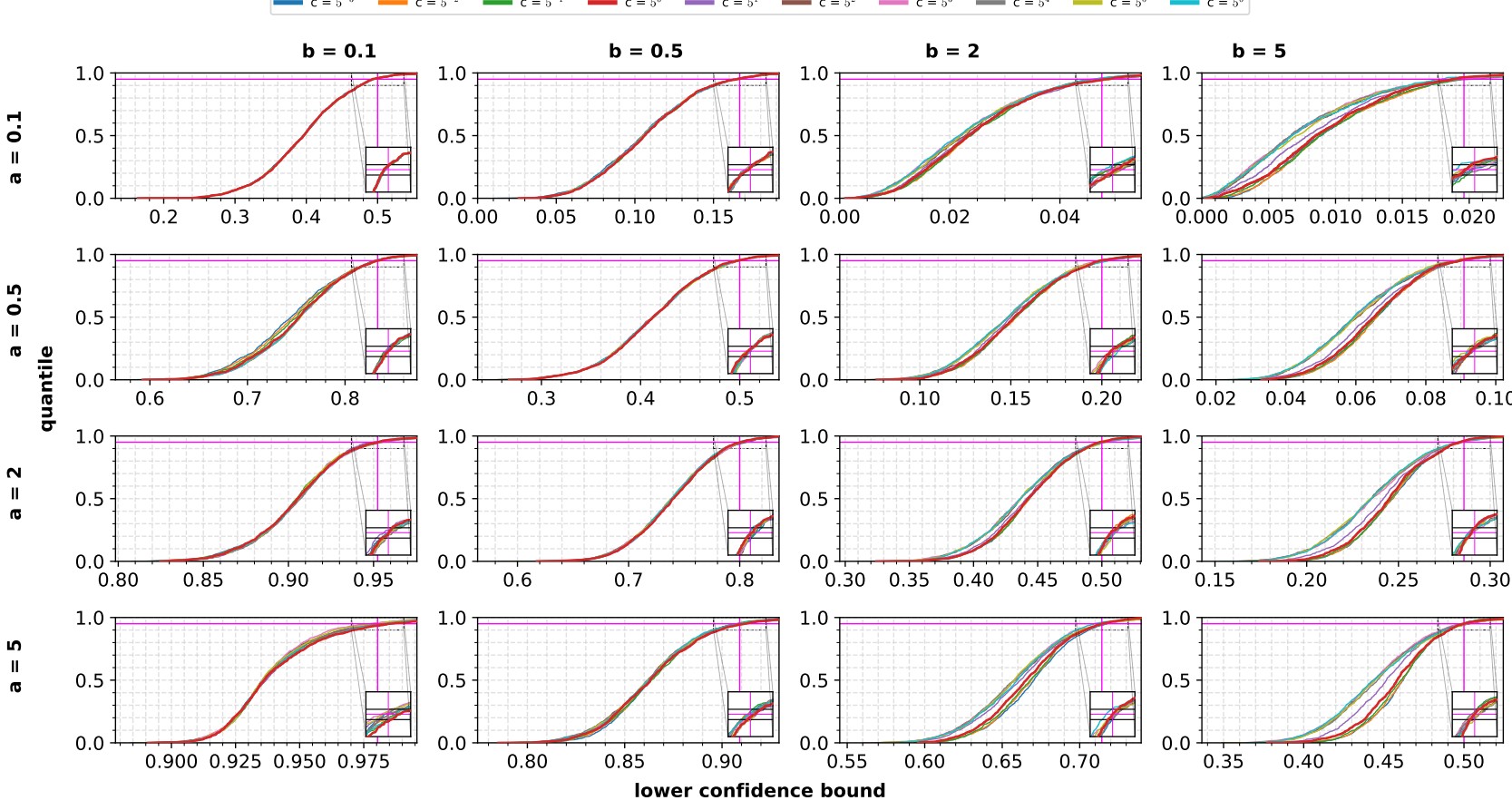

### C.4 Last round bet

Here we provide experiments quantifying the effect of (an additional) last round betting described in Appendix D.2.

- STaR-Bets is Algorithm 4 with detail from D.
- Bets is STaR-Bets without the ✸-component.
- (STaR) bets w/o last bet are the first two algorithms without the last round bet (described in Appendix D.2).
- Hedged-CI is the confidence interval of [23].

We can see that Hedged-CI performs similarly to Bets without last bet, and STaR bets without the last bet is significantly stronger. Adding last bet help both algorithms, but the effect is less significant on STaR, since by design, it tries to end up with $0$ or $\frac{1}{\delta}$ money. Sometimes the performance of Hedged-CI and Bets without last bet (and also of the pair STaR with/without last bet) are so similar, that the curves are indistinguishable.

**Bernoulli, delta = 0.05, averaged over 1000 runs**

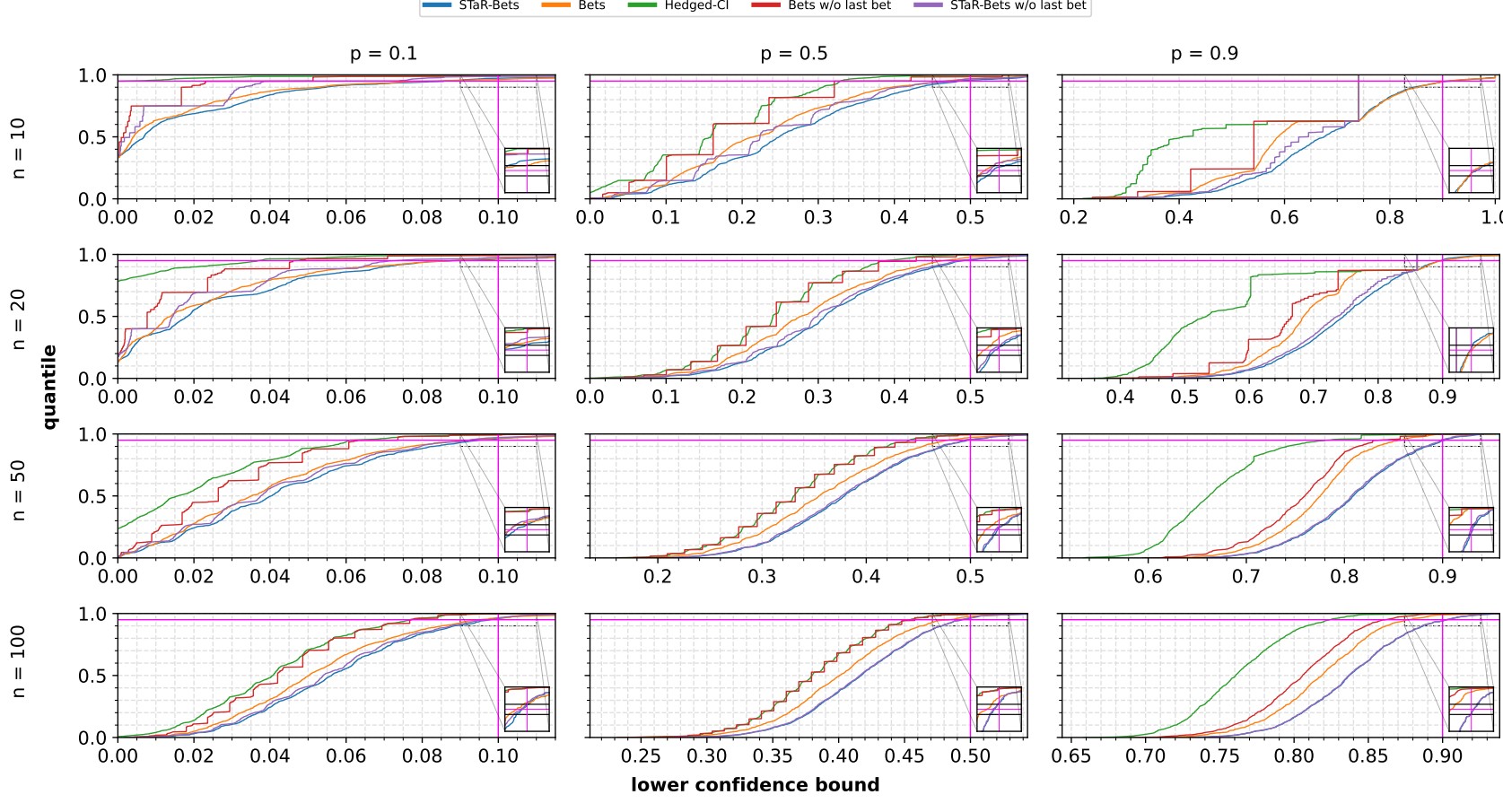

**Beta, n = 100, delta = 0.05, averaged over 1000 runs**

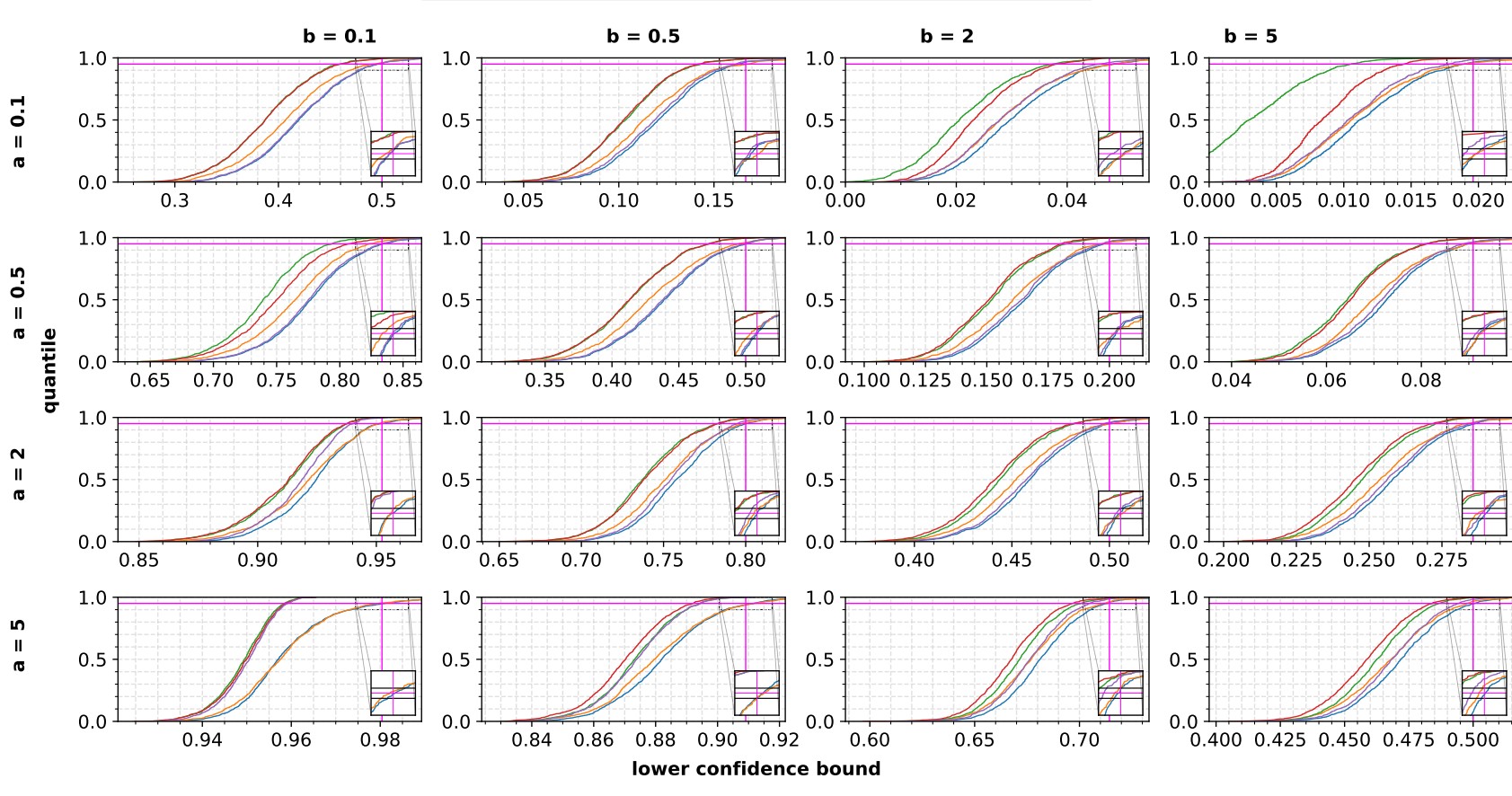

**Beta, n = 1000, delta = 0.05, averaged over 1000 runs**

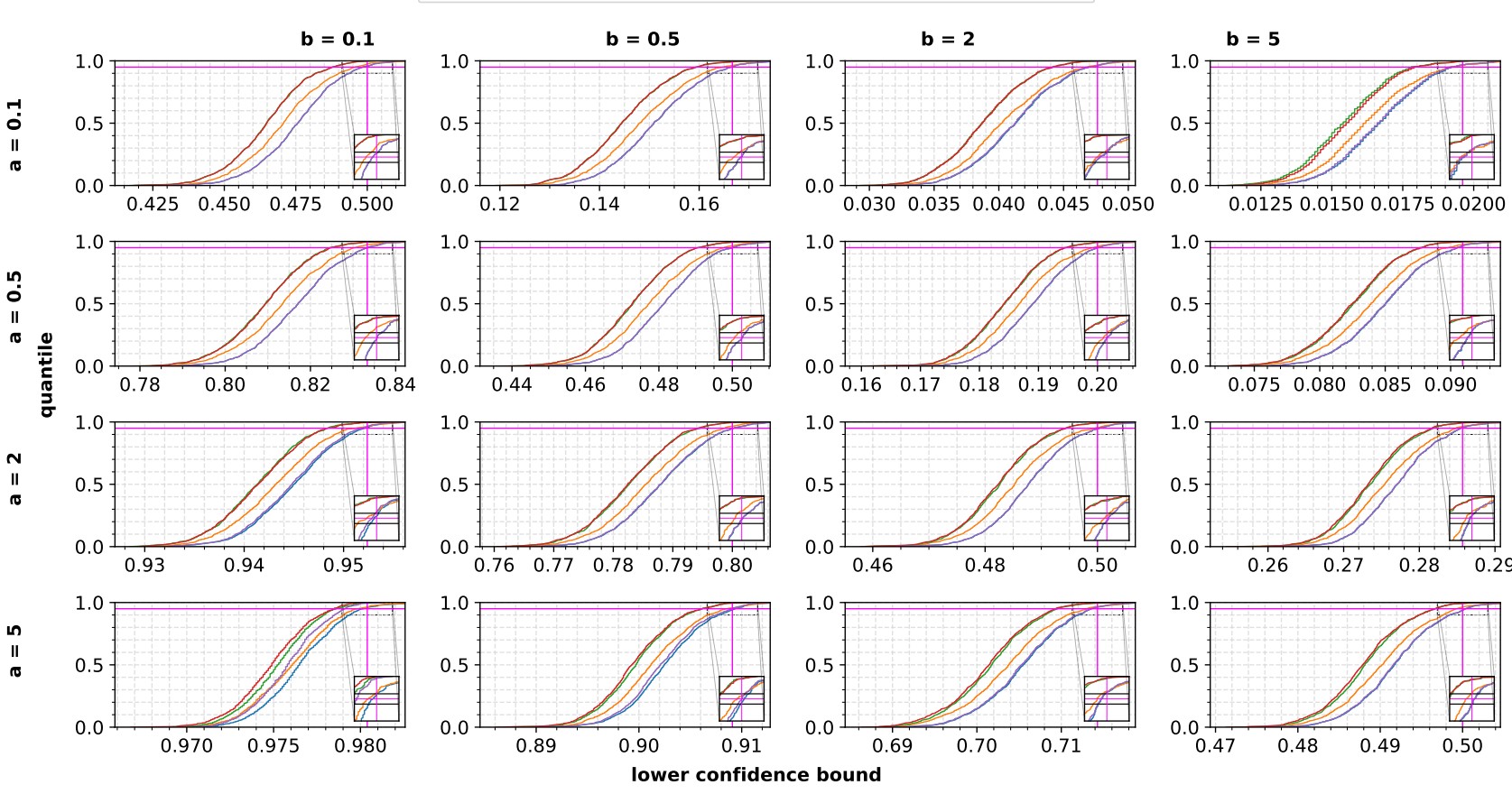

## C.5 Variance of the confidence interval with data shuffling

While the majority of confidence intervals are independent of the data order, it is not the case for the betting based one. Here, we present several experiments where for every setting, the corresponding sample of random variables was obtained and then it was only shuffled for all the 1000 experiments. Note that here the coverage is meaningless, as we do not draw fresh samples.

**Bernoulli, delta = 0.05, averaged over 1000 runs**

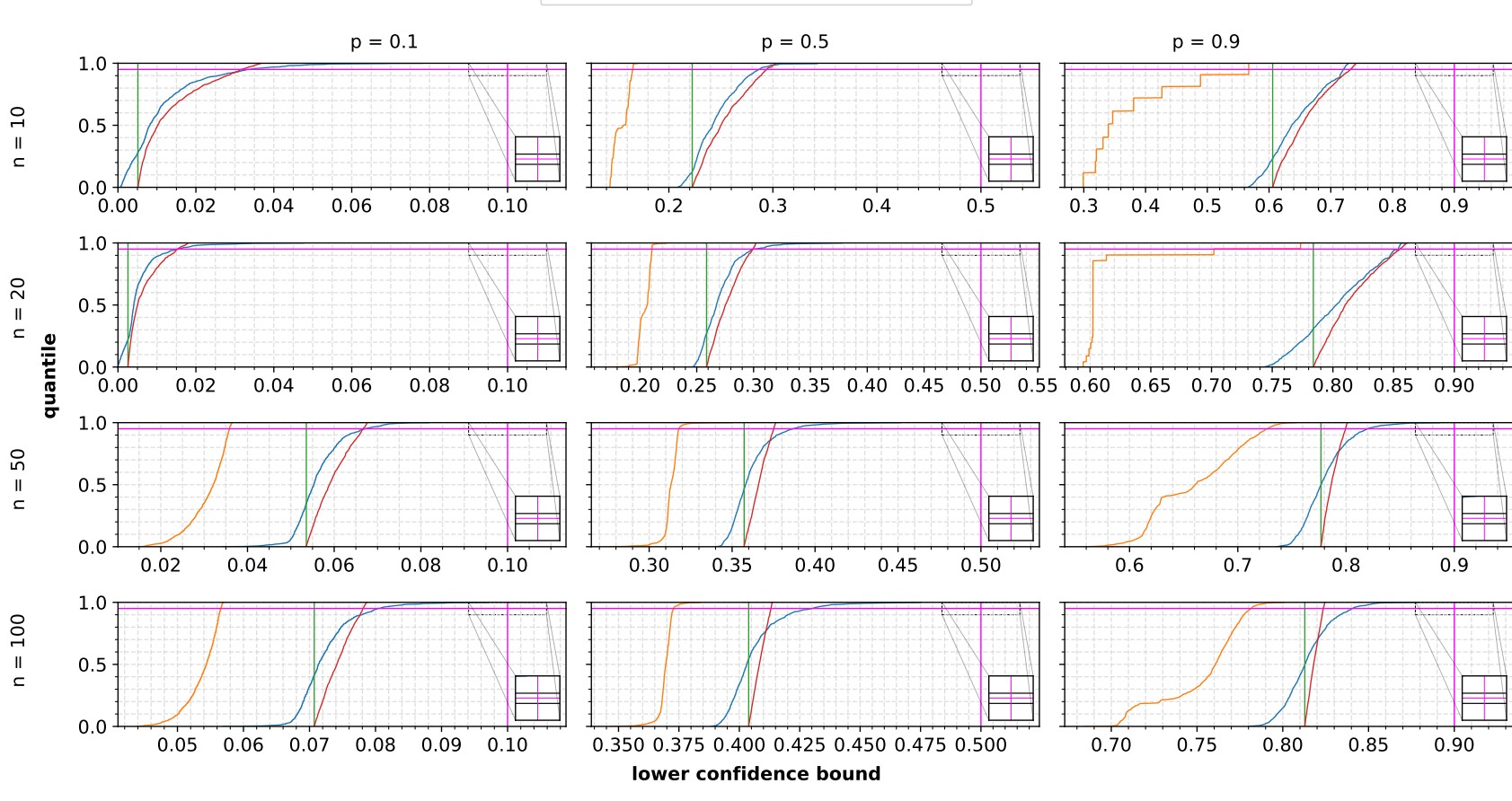

**Beta,  n = 50,  delta = 0.05, averaged over 1000 runs**

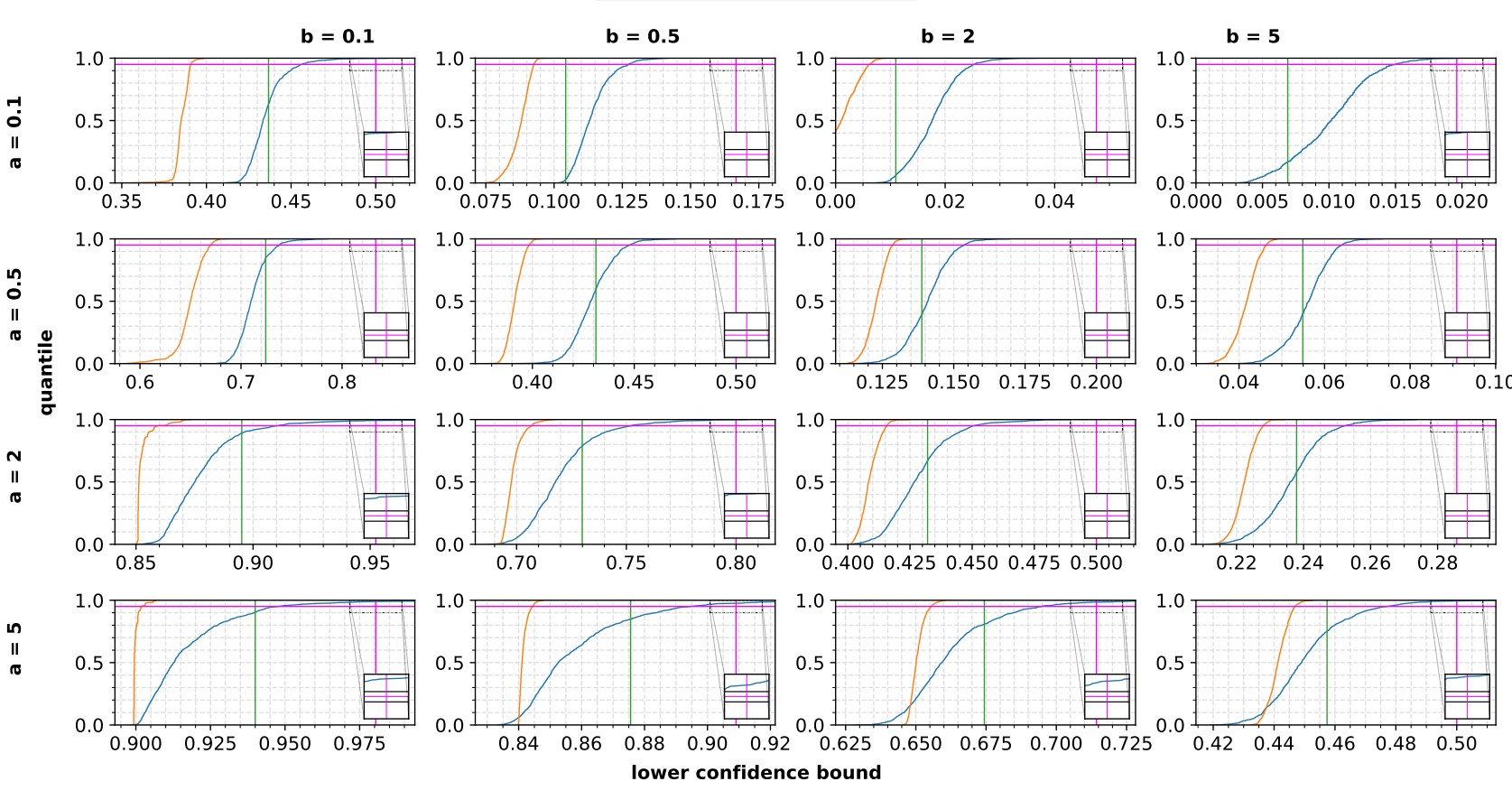

# D  Implementation details

There are certain aspects where the algorithm we implemented deviates from the vanilla version in Algorithm 3, but the underlying process is still a Test process, so the resulting confidence intervals are valid. Here, we describe these details.

## D.1  Clipping

Instead of clipping the estimate of $\mathbb{E}[(X-m)^2]$ to its upper bound $\max\{m^2, (1-m)^2\}$, we clip it to $m(1-m)$ instead. It would be the maximal value of $\mathbb{E}[(X-m)^2]$ if $m = \mathbb{E}[X]$. The testing problem is hard when $m \approx \mathbb{E}[X]$ (and easy otherwise), so this way, we help the algorithm on the hard values of $m$ and hurt it on the easy ones, yielding better empirical performance.

## D.2  Last round randomization

Just as the conservativeness of Clopper-Pearson is fixed by randomization, we attempt similar thing in the betting games.[5] After finishing the testing procedure, we end up with wealth W. Then, we draw a uniform random variable $U$ supported in $[0,1]$; if $W \geqslant U/\delta$, we set it to $\frac{1}{\delta}$. Otherwise, we set it to $0$. It is easy to see that after this manipulation, $W$ is still a test-variable, since it is still non-negative and its expectation did not increase. The effect of this is usually small for ✶-Bets, since it is encouraged by design to end up with $1/\delta$ or no wealth. The exceptions are instances with small $n$, small variance, or instances with $\mathbb{E}[X] \sim 1$, in which cases the bets has to be very small, and so it is not easy to properly adapt the betting strategy and ensure that we bet aggressively enough if needed. See Appendix C.4 for experiments.

## D.3  Choice of second moment estimator

Algorithms 3, 4 have a hyperparameter $\alpha$ corresponding to the lower bound of probability of having short intervals in the proofs. It influences how conservative we should be in estimating $\mathbb{E}[(X-m)^2]$. We have (empirically) observed that the larger $m$ is, the more conservative we should be. The intuition is that our win (or loss) $\ell(X-m)$ can be as low as $-\ell m$, and so with larger $m$, we should be more careful about the choice of $\ell$. We instantiate the $10 \log \frac{8}{\alpha} n/(t-1)^2$ term as $cmn/(t-1)^2$ with $c = 1$. In Appendix C.3 we provide experiments suggesting that the exact choice of $c$ is not that crucial, but a proper argument about how should $c$ depend on $m$ is not given.

---

[5]The resulting random variable is known as all-or-nothing random variable, see [17].

