# OpenReview forum: "STAR-Bets: Sequential TArget-Recalculating Bets for Tighter Confidence Intervals"
_NeurIPS.cc/2025/Conference — NeurIPS 2025 poster_

### Official Review · Reviewer_Epqr · 2025-07-02

**Clarity:** 2
**Significance:** 3
**Originality:** 3
**Rating:** 3
**Confidence:** 4

**Summary:**

This paper proposes a method for constructing fixed-time, non-asymptotic confidence intervals through a testing-by-betting approach similar to the work of [1]. The authors leverage a general framework called Sequential Target-Recalculating to size their bets to ensure that the "wealth" of their process aims to reach the desired $1/\delta$ level by the fixed time of inference $N$. The authors leverage their framework to show that their approach strictly improves classical confidence intervals based on Hoeffding and Bernstein inequalities, and provide empirical results demonstrating the efficacy of their methods.

[1] I. Waudby-Smith and A. Ramdas. Estimating means of bounded random variables by betting. Journal of the Royal Statistical Society Series B: Statistical Methodology, 86(1):1–27, 2024.

**Questions:**

* The Markov inequality stated in Proposition 1 provides error guarantees at a fixed time $N$; the related martingale concentration equality (Ville's inequality [2]) uses the same threshold $1/\delta$, while providing time uniform guarantees. Is there a reason as to why we only construct confidence intervals at time $n$, rather than at any arbitrary times $t \leq n$?
* Is there a particular reason for this approach applying only to bounded means? Sequential testing methods often generalize to other settings, such as when variables have bounds on their MGF [3].
* The test process provided in Defn. 1 is strikingly similar to the definition of $e$-processes [1] in the any-time valid literature. As I understand this work, the STaR method is a tailored approach for fixed sample testing that essentially "bets all wealth" before the final time $n$, ensuring that the conclusion of reject / accept is achieved by time $n$. Does this approach generalize to existing testing-by-betting methods, such as Hedged?

[2] Jean Ville. E ́tude critique de la notion de collectif. Gauthier-Villars Paris, 1939
[3] https://arxiv.org/abs/1810.08240

**Ethical Concerns:**

["NO or VERY MINOR ethics concerns only"]

**Final Justification:**

Despite the novelty of the approach (tuning a betting scheme to be tight given knowledge of the horizon), this work is ultimately limited by a lack of examples beyond the bounded mean setting, which has been extensively studied in the literature. The ideas of this work are sound and solid (which is why we maintain a score close to acceptance), but this work could benefit from being framed more generally as an approach for tuning anytime valid methods for a fixed horizon.

**Limitations:**

* The results of this work may be applicable to many testing-by-betting and anytime valid methods as a way to covert anytime-valid $e$-processes to fixed-time, non asymptotic confidence intervals. However, the authors focus specifically on bounded-mean confidence intervals. I would be happy to improve my score if the authors provide intuition on this framing.
*  The authors miss the opportunity to state anytime-valid guarantees for their test, which they implicitly use in their main algorithm (Algorithm 4).
* The authors may be missing existing works that leverage similar principles for generalizing a fixed-time approaches to a similar framework. [4] proposes a generalization of the standard $t$-test to an anytime valid test, and has a similar update (where the denominator of the sequential term at time $t$ is divided by $O(n-t)$).


[4] https://arxiv.org/pdf/2501.03982

**Paper Formatting Concerns:**

I have no concerns on formatting.

**Quality:**

2

**Strengths And Weaknesses:**

**Strengths**
* The approach proposed by the authors provides a flexible framework for improving existing concentration inequalities for use as confidence intervals. The authors demonstrate this by applying their framework to well-known concentration inequalities such as Hoeffding or Bernstein.
* Empirically, the authors provide extensive baseline comparisons for means of bounded random variables, including specialized confidence intervals for the Bernoulli case.

**Weaknesses**
* Some sections of this paper may be unclear to the reader: for example, the scenarios in pg. 3 of the paper are not quite clear and only seem somewhat related to the aims of the paper.
* As opposed to a commonly used baseline (Hedged), the confidence intervals proposed in this work are only valid at a fixed time. It would be more fair for authors to compare fixed-time confidence intervals to similar methods, as opposed to anytime-valid approaches that are necessary wider than fixed-time inference due to protecting type I error at all times, rather than just a single sample size $N$.

---

> ### Author Rebuttal · Authors · 2025-07-28
>
> We thank the Reviewer for their feedback.
> We agree that our writing could be improved, but this is an easy change that we can do in the camera ready version.
> Moreover, given that the only listed weaknesses are the improvable writing and the misunderstanding of the unfair comparison, we would be grateful if the Reviewer could reconsider their score as foreshadowed in the review.
>
> We answer the specific questions in the following.
>
>
> ## Weaknesses
> **Scenarios might be unclear and only seem somewhat related to the aims of the paper**\
> We present several examples of betting games and strategies from which we made some observations that motivate our proposed betting strategy. We could have just stated our algorithm, but we feel it is better to present intuitions on why one can improve over a constant betting strategy. Notably, Reviewer QY1z mentioned that they specifically liked this part.
>
> **Unfair comparison with Hedged CS**\
> This is not correct: The comparison is completely fair because we compare with the fixed time version of Hedged (Remark 3 of [19]), called Hedged CI. The confusion of the Reviewer is probably due the fact that [19] presents two Hedged algorithms: Hedged-CI (fixed time) and Hedged-CS (time-uniform).
>
> ## Questions
>
> **Why not intervals at all times t<n?**\
> As you note, we can (trivially) do this and compute the "current" interval after every observed sample at no additional cost with the same final interval. However, we explicitly decided not to do it because our sole focus is on CIs and not CSs, and we did not want to confuse the readers. When one needs a CS, we would not recommend our algorithm.
>
> **Why only bounded means? (we assume this is a typo and should have been means of bounded random variables)**\
> The class of bounded random variables is a rich and practical regime (it is rather easy to check if my random variable is bounded, rather than checking if it has (say) c-subgaussian tails).
> It is probably possible to extend this idea to other classes of confidence intervals (the linked references are for confidence sequences, so it cannot be directly applied here).
>
> **Does STaR generalize to hedged?**\
> As we write above, there are hedged algorithms: CI and CS. Hedged CI differs only mildly (in the variance estimator, for the sake of proof) from our "base" algorithm (cf, lines 281-284) and the STaR technique can be directly applied to it; one would get similar empirical results to ours. On the other hand, the STaR technique is not applicable to CSs (there is no end time when we want to have either 1/delta or 0 money).
>
>
>
>
>
>
> ## Limitations
>
> **The results of this work may be applicable to many testing-by-betting and anytime valid methods as a way to covert anytime-valid -processes to fixed-time, non asymptotic confidence intervals. However, the authors focus specifically on bounded-mean confidence intervals. I would be happy to improve my score if the authors provide intuition on this framing.**\
> As the Reviewers note, our approach might have a larger number of applications. In this paper, we decided to focus on the problem of confidence intervals for bounded random variables because this is a widely known setting that is both foundational and important from a practical point of view.
> As a matter of fact, we started working on this topic because applied people explicitly asked us for a better method for the calculation of valid confidence intervals of bounded random variables for deep learning applications, as, for example, in
>
> Angelopoulos AN, Bates S, Fannjiang C, Jordan MI, Zrnic T, Prediction-powered inference, Science, 2023
>
> In future work, we will explore the application of the STaR method to other settings in the testing-by-betting literature.
>
>
>
> **Anytime guarantees missing**\
> As we wrote above, it is trivial to get the anytime-version. We decided to not include it for the sake of clarity.
>
> **Comparison with [4]**\
> Thank you for pointing out the work of Koning & van Meer (2025). They indeed have a similar n-t term, but the connection to our work is only apparent. More in detail, their method does not improve the test at time n, instead they want to (potentially) provide some intermediate rejections at times t<n. This is not at all our objective. Instead, we aim at improving the rejection rate at t=n. Hence, their paper and our work are orthogonal. We will add a discussion on this point to avoid confusions between the two methods.

---

> > ### Comment · Reviewer_Epqr · 2025-08-06
> >
> > To note some points regarding the baselines:
> > * The Hedged CI in Remark 3 of [19] is indeed anytime valid - the choice of $\lambda$ they pick for their fixed-time confidence interval is particularly tight at the final sample size $n$, but it maintains its anytime-valid guarantees by being supermartingale (as long as $\lambda$ is bounded above zero by a constant) under the null $\mu = m$. Thus, even while [19] presents its approach as a "fixed-sample" test, it actually retains its anytime-validity property due to being a supermartingale. This is why they are able to do repeated intersections of their confidence interval across the horizon.
> >
> > Given the authors' aims and scope for this work, I will leave my rating as is - I believe the manuscript could benefit from additional clarifications regarding applications to other types of finite CIs and distinctions from anytime-valid methods.

---

> > > ### Author Response · Authors · 2025-08-06
> > >
> > > **Hedged-CI retains its anytime-validity property due to being a supermartingale. This is why they are able to do repeated intersections of their confidence interval across the horizon.**
> > >
> > > Could the Reviewer please tell us how the comparison is unfair, when they have written in one of the questions that our algorithm has the same property they have used in the argument now?
> > > In particular, the property above holds in our algorithm too: we also produce a supermartingale and we could, as you correctly observed, produce confidence intervals for each time step. Finally, our method is also designed to produce the tightest CI after $n$ observations, as theirs. Note that taking intersections of intervals in our case would not change the result in any way, because our method stops betting when the threshold to discard a testing value of the mean is crossed.
> > > That said, Hedged-CI is still the SotA method for CI for bounded random variables and this is the only thing that matters in the evaluation.
> > >
> > >
> > > **Given the authors' aims and scope for this work, I will leave my rating as is - I believe the manuscript could benefit from additional clarifications regarding applications to other types of finite CIs and distinctions from anytime-valid methods.**
> > >
> > > We are not sure what the Reviewer means here: We answered all the concerns of the Reviewer and we clarified the connection with other work. Is there anything missing or wrong in our answers? If this is not the case, your rejection appears to be solely based on the fact that our method is for bounded random variables. *Given that this is a machine learning conference and given that there is prior work on this very topic, we respectfully but very strongly disagree that this might be considered a valid reason to reject any paper and we explicitly call the attention of the AC on this point.*

---

> > > > ### Comment · Reviewer_Epqr · 2025-08-06
> > > >
> > > > To answer the first question, the Hedged approach presented in Remark 3 of [19] is anytime-valid for all $t \in \mathbb{N}$ - this means that it inherently has spread the error tolerance budget $\delta$ across an **infinite** potential time horizon. In contrast, because this method bets all wealth by the time $n$, the error budget is only spread across all $t \leq n$, where $n$ is the pre-specified fixed time for inference. The algorithm proposed here is anytime-valid up for the horizon $t \in \{1,..., n\}$, while the baseline is anytime-valid for the horizon $t \in \mathbb{N}$, which inherently makes it a weaker baseline.
> > > >
> > > > I do not mean to antagonize the author with the last comment - apologies if it read as poorly worded. I do believe that this work poses novel ideas, but I think a reframing of the core contribution (which is the targeted recalculation of bets with explicit knowledge of the horizon length) could be presented in a different way (such as a general-purpose approach for any CI) that makes this contribution overall stronger as a submission.
> > > >
> > > > I do appreciate the author's clarifications, and I will consider this feedback into my final score.

---

> ### Author Response · Authors · 2025-08-07
>
> We thank the Reviewer for clarifying their view. Here, we just want to explain one subtle point.
>
> **This method bets all wealth by the time $n$, so the error budget is only spread across all $t \leq n$**
>
> This is not completely exact: our algorithm bets more aggressively approaching $n$, but this does not imply that by the time we reach $n$ the wealth is 0. In fact, even betting everything, to lose all the money we should observe a value of the random variable of 0 or 1 (depending on the bet). Hence, unless the random variable has probability mass exactly on 0 and 1, in practice the algorithm will likely have available wealth in all means to be tested. So, if needed, one can keep using the algorithm after $n$, just setting the betting strategy to anything else after $n$. In this view, Hedged-CI and our algorithm are not fundamentally different in how they use with the budget $\delta$ over the infinite number of rounds. The only real difference is that our algorithm *will try its best to use its budget in the first $n$ rounds*, that directly results in tighter intervals.

---

### Official Review · Reviewer_B2su · 2025-07-03

**Clarity:** 2
**Significance:** 3
**Originality:** 2
**Rating:** 2
**Confidence:** 3

**Summary:**

This work considers the problem of assigning tight confidence intervals to the mean of a finite-size sample from a given distribution. To that end, a sequential betting-based algorithm is constructed with finite-sample optimality guarantees and performance gains over the existing non-sequential algorithms.

**Questions:**

1. Since future samples cannot be affected by any action by the observer, there is no inherent advantage in constructing CIs sequentially. The only reason I could think of for working sequentially is data spoofing. Do you have a more ethical reason for doing this?

**Ethical Concerns:**

["NO or VERY MINOR ethics concerns only"]

**Final Justification:**

There is no underlying reason for working sequentially except for data spoofing. The paper is poorly written both in terms of content and in terms of grammar.

**Limitations:**

No. Please my comments and questions above.

**Paper Formatting Concerns:**

I did not spot any formatting concerns.

**Quality:**

2

**Strengths And Weaknesses:**

The problem of designing tight confidence intervals (CIs) for finite-size samples is important.

The following issues are of concern:
* The paper contains multiple English mistakes. In some cases, these errors hinder clear understanding.
* Bootstrapping algorithms, which are widely used by statisticians when the underlying distribution is unknown, are entirely overlooked.
* The authors seem to be interested in determining a tight CI at a specific horizon $n$. As no "active control" over the samples is possible in this setting (in contrast to, say, reinforcement learning), sequential algorithms do not possess any advantage over non-sequential "batch" algorithms. However, the lengthy exposition about betting and lines 171-173 implicitly suggest the opposite. In fact, lines 171-173 seem to suggest that more data can deteriorate performance.

Further comments:
* Def. 1: The last restriction means that W is a supermartingale. Please state this. Also, $M_0$ should be $W_0$.
* Theorem 2: This theorem is not stated clearly. In particular, $S$ is not unique, as various CIs may be constructed to meet a certain confidence level.
* There: In the "testing by betting" part, no relation between $W$ and $X$ is mentioned.
* Line 145-146: The inline equation seems to be at odds with the fact that $W$ is a supermartingale.
* Sec. 3: This example ( and "Scenarios") should be explained in the context of CI assignment. Also, please make the reference to this example in lines 147-148 clearer.

---

> ### Author Rebuttal · Authors · 2025-07-28
>
> We thank the Reviewer for their feedback.
> In the following we answer their specific concerns.
>
> ## Weaknesses
> **Bootstrapping algorithms are overlooked**\
> We (and apparently previous work on this topic too) are not aware of bootstrapping algorithms for confidence intervals with guaranteed coverages. That said, we would be happy to be corrected by the Reviewer on this point.
>
> **The problem setup is never rigorously defined … In particular, are the samples $X_1, ..., X_n$ assumed i.i.d.?**\
> Actually, it is defined: Please see the sentence starting at the 6th line of the paper (line 25) starting “So, more formally, after observing $n$ i.i.d.  samples $X_1, \dots , X_n$ ..”
>
> **Sequential algorithms do not possess any advantage over non-sequential "batch" algorithms. However  (lines 171-173)  suggest the opposite**\
> While there is no advantage in general, lines 171-173 comment on the fact that the derivation of standard (“batch”) concentration inequalities, as Hoeffding, Bennett, and Bernstein, can be interpreted as sequential betting schemes where one selects a constant bet beforehand. In these cases, we provably show that this can be improved by recalculating the best bet before every observation and this is the STaR technique.
>
>
> ### Further comments
> * A typo; thanks.
> * **Thm 2 is not stated clearly** Actually, Theorem 2 is stated clearly: It is a scheme, so you can choose the individual components to arrive at different confidence intervals. All the confidence intervals we derived in the paper (Hoeffding, Bernstein, all our new algorithms) are correct by Theorem 2.
> * **relation W on X is not mentioned** The relation can be arbitrary (even none), it is only important that W is a test process under the null hypothesis.
> * **inline equation  at odds with the fact that W is a supermartingale** It is actually correct because here we comment on a specific test process (the display just before) which happens to be a martingale, thus the equality.
>
>
> ## Questions
> **What are the model assumptions and exact problem formulation?**\
> See first paragraph of the paper. Observe i.i.d. sample $X_1, ... X_n$ and produce confidence interval for the mean parameter. Then we want to produce as tight (short) confidence interval as possible.
>
>
> **Why don’t you start with the easier batch setting and rather do it sequentially?**\
> We believe the Reviewer might have the wrong intuition on this problem.
> In fact, in Section 4.1 we explicitly prove that the sequential approach is never worse and it can be better than the batch approach (see Corollary 5). Let’s try to give some intuition on why this is the case, for example considering the Bernstein bound (Appendix B). In the Bernstein case, we know that the optimal asymptotic bet (that is, the “batch” bet) depends on the variance of the random variable. However, *this kind of bet does not take into account what actually happened on a specific sequence*, that is, it is independent of the outcomes. Instead, by sequentially recalculating the optimal bet, we take into account the past outcomes, through their effect on the wealth. At this point the Reviewer might wonder why only using the past to calculate a bet and not the future too? This would not work, because we still need to generate a supermartingale.
> Finally, we urge the Reviewer to focus on the fact 1) the previous SotA results were also from a sequential method (Hedged-CI) and 2) our approach gives SotA results on this problem.

---

### Official Review · Reviewer_Toih · 2025-07-05

**Clarity:** 2
**Significance:** 2
**Originality:** 2
**Rating:** 3
**Confidence:** 4

**Summary:**

This paper considers the problem of constructing confidence interval for the mean of bounded random variable, given a set of i.i.d. observations from the same distribution. The author uses the betting process to construct the confidence interval, and propose the "$\ast$-technique" to improve on the current betting method. Specifically, the "$\ast$-technique" is to substitute $\mathrm{log}(1/\delta)$ and $n$ with $\mathrm{log}(1/\delta W)$ and $n-t+1$ during the betting process, leading to a more adaptive process. Experiments were conducted to compare the proposed method with other state-of-the-arts, and shows a promising empirical advantage of the proposed method, STaR-Bets.

**Questions:**

Writing:

1. The paper is relatively easy to follow, though it would be better if there is a paragraph summarizing the organization of the paper.

2. For the related work section, it would help significantly if the author can provide a table that summarizes the theoretical guarantees (e.g., rates or forms of optimality) for each method discussed. It would greatly help clarify the landscape. At present, the discussion is purely verbal, and the claims regarding optimality remain somewhat vague.

3. Related to the point above, in the experiment section, when comparing different methods, could the authors write out each method more explicitly (maybe in the appendix)? Also, why does Figure 2 present two plots with different numbers of methods shown?


Theoretical results related:

1. Typo from Definition 1 & Proposition 1: I think the author intended to define the test process as a martingale, so in definition 1, the last $\le$ should be $=$ instead.

2. Incoherent argument in the proof of Theorem 2: Please double check the proof. For example, the last sentence, $1-\delta$ should be $\delta$.

3. Unclear notation from the proof of Proposition 4: The author define $l^{H} = argmin_{\lambda \in \mathcal{C}} \sum_{i=1}^{n} X_{i}$. However, it is unclear how does the optimization objective related to $\lambda$?

4. Interpretation of Theorem 8: Could the author elaborate a bit more on how this result shows the optimal width? How does it rely on the knowledge of the variance $\sigma^2$?

**Ethical Concerns:**

["NO or VERY MINOR ethics concerns only"]

**Limitations:**

An in-depth comparison on the theoretical properties between the proposed method and existing methods is lacking.

**Paper Formatting Concerns:**

No any major formatting issues.

**Quality:**

2

**Strengths And Weaknesses:**

Strength:

1. The idea of the "$\ast$-technique" is interesting and intuitive, since finding a way to make the betting process more adaptive is crucial to improve CI’s informativeness.

2. Empirical performance of the proposed method is promising compared to other state-of-the-art methods.

Weakness:

1. The writing of the paper should be improved to enhance clarity. Specific suggestions and questions are listed in the Questions section below.

2. Given that confidence interval construction for the mean under i.i.d. data is a classical and well-studied topic, it is not clear what substantive new theoretical insights this work contributes beyond existing methods. The paper would benefit from clarifying the practical and theoretical advantages of the proposed approach. Please see more concrete questions and concerns listed in the Questions section below.

---

> ### Author Rebuttal · Authors · 2025-07-28
>
> We thank the Reviewer for their feedback. In the following we address their comments and questions.
>
> ## Weaknesses
> **This is a classical and well-studied topic, it is not clear what substantive new theoretical insights this work contributes beyond existing methods.**\
> We proposed a method producing significantly shorter confidence intervals than state of the art (see Appendix C, we performed and condensed (literally) millions of experiments) with significant evidence that further improvements are unlikely to be significant. On the theory side, we provided the first analysis of a betting based confidence interval reaching the optimal width (up to a constant factor).
>
> ## Questions
>
> **For the related work section, it would help significantly if the author can provide a table that summarizes the theoretical guarantees (e.g., rates or forms of optimality) for each method discussed. It would greatly help clarify the landscape. At present, the discussion is purely verbal, and the claims regarding optimality remain somewhat vague.**\
> In general, in the setting of bounded random variables the optimal rates are known and attained. However, the “optimal” methods are known to perform poorly on small sample sizes. This has motivated the line of research of confidence intervals and confidence sequences based on betting algorithms, because they give rise to much shorter intervals.
> Before our paper, the SotA method was by Waudby-Smith and Ramdas (2024) (247 citations). We establish a new SotA by beating their method both empirically (see experiments) and theoretically (see Theorem 8, whereas the methods in Waudby-Smith and Ramdas (2024) have only an asymptotic convergence without a rate).
>
>
> **Related to the point above, in the experiment section, when comparing different methods, could the authors write out each method more explicitly (maybe in the appendix)?**\
> Good point, we can definitely expand the discussion of the competing methods in the Appendix.
>
> **Why does Figure 2 present two plots with different numbers of methods shown?**\
> Figure 2 presents results for two different random variables (see titles): Beta(5,1) and Bernoulli(0.3). Note that the Clopper-Pearson interval is applicable only to Bernoulli (binomial) random variables, thus is not present in the plot with the Beta distribution experiments.
>
> **Typo in definition 1, should be martingale**\
> No, we are happy to include supermartingales (yielding Hoeffding and Bernstein inequalities) as indicated by footnote 2.
>
> **Typo in proof of Theorem 2**\
> Yes, thank you, we will fix.
>
> **Unclear notation from the proof of Proposition 4: The author define $l^{H} = argmin_{\lambda \in \mathcal{C}} \sum_{i=1}^{n} X_{i}$. However, it is unclear how does the optimization objective related to $\lambda$?**\
> We are sorry for the (understandable) confusion, we will fix this. The set of feasible lambdas $C$ depends on $\sum_i X_i$ and it should have been a condition rather than a constraint set. $\ell^H$ is the argmin over $\lambda$ of the following optimization problem ($m,n,\delta$ are constants):
> $$
> \min_{X_1, \dots X_n \in [0,1], \lambda \in \mathbb{R}} \sum_{i=1}^n X_i, \quad
> \text{s.t.}\quad  \lambda\sum_{i=1}^n (X_i-m) -n\lambda^2/8 \geq \log\frac1\delta
> $$
>
>
>
> **Interpretation of Theorem 8: Could the author elaborate a bit more on how this result shows the optimal width? How does it rely on the knowledge of the variance $\sigma^2$?**\
> The result shows that whenever you want a confidence interval of the optimal width up to a (1+c) factor for c >0, there is always a number of samples (depending on the variance and other things) such that the algorithm produces a confidence interval of the optimal width up to the given factor. To determine the minimal number of samples needed, we must know the variance, but the algorithm is running without this knowledge. This is necessary, since the minimal width of confidence interval is $\log(1/\delta)/n$ and we need $\sigma^2 > \log(1/\delta)/n$ for the asymptotical regime to kick in. (Also, it is easy to show that $n=O(c^{-8})$ when treating everything as else as constants).
>
> ## Limitations
> **An in-depth comparison on the theoretical properties between the proposed method and existing methods is lacking.**\
> As said above, optimality in this setting is easy. Instead, one strives for numerically small intervals *and* theoretical guarantees. The only other algorithms to satisfy these constraints are the algorithms in Waudby-Smith and Ramdas (2024). They propose to use heuristic strategies that only guarantees an asymptotic convergence to the optimal width of the confidence interval, but no explicit rate. Note that the absence of rate might not be fixable because their estimator of the variance might converge arbitrarily slowly on some distributions. Instead, we prove a finite time rate for our approach. On this point, it is also easy to calculate that $c$ in Theorem 8 decays as $n^{-1/8}$ (this treats all the other problem parameters as constants when we only increase $n$) when selecting constants in the proof as In Appendix A (lines 405-408).

---

### Official Review · Reviewer_QY1z · 2025-07-06

**Clarity:** 2
**Significance:** 4
**Originality:** 4
**Rating:** 6
**Confidence:** 3

**Summary:**

The paper considers the problem of constructing confidence intervals for the mean of a bounded random variable, given access to a fixed number of samples, and proposes new methods for constructing confidence intervals using betting algorithms. Previous betting-based confidence intervals are either designed for anytime-validity, and as a result are sub-optimal for a fixed sample size, or lack non-asymptotic bounds on their width. The main idea behind the proposed star-betting method, is to design betting algorithms that always finish with either 0 wealth or just enough wealth to reject a candidate value for the mean, as opposed to betting algorithms that simply try to achieve maximal wealth. This property is shown to yield confidence intervals with exact coverage. In addition, the testing by betting method of Waudby-Smith and Ramdas is modified such that it produces confidence intervals whose widths satisfy an optimal non-asymptotic bound.

**Questions:**

I’m a bit puzzled by what $\ell^H$ in the proof of Proposition 4 is supposed to be. It’s defined as the argmin of $\sum_{i=1}^{n}X_i$, w.r.t. $\lambda$ in some set. Does this mean $\ell^H$ is any feasible value? $\ell_{\star}^{H}$ is defined to be the minimiser of $\sum_{i=t}^{n}X_i$. Is this also just any feasible value?

In Theorem 8, is it possible to say how quickly the constant c decays as n increases?

The following is a list of suggested fixes to typos or weird sentences:
- Line 27: “optimal asymptotically properties” to “optimal asymptotic properties”
- Line 36 “testing martingale” to “test martingale”
- Line 40 “fail short” to “fall short”
- Line 43 “betting algorithm” to “betting algorithms”
- Line 51 “Bernestein’s” to “Bernstein’s”
- Line 85 “a betting algorithm does” to “a betting algorithm makes”
- Line 131 I think M_0 should be W_0
- Line 135 Given the definition of a test process, there should be an inequality $\mathbb{E}[W_n] \leq \mathbb{E}[W_0]$ and a non-strict inequality $W_n \geq 0$.
- Line 146 “we have the non-negativity” to “$W_i^m$ is non-negative”
- Line 150 “such approach” to “such an approach”
- Line 157 “monotonous” to “monotonic”
- Line 203 “And on the other hand” to “On the other hand”
- Line 204 “Resulting one bankruptcy” to “resulting in bankruptcy”
- Line 223 $W_{i=1}$
- Line 226 “, but briefly, we will motivate it” should this be “, but later, we will motivate it more formally”?
- Line 267 “the $\star$ version of just derived Algorithm 3” to “the $\star$ version of Algorithm 3”
- Line 276 “with T-test” to “with the T-test”
- Line 285 “and empirical Bernstein bound” to “and the empirical Bernstein bound”
- Line 286 “Empirical Bernstein bound” to “The empirical Bernstein bound”
- Line 289 “Method of Phan et al.” to “The method of Phan et al.”
- Line 293 “how do the widths” to “how the widths”
- Line 304 “How much does it helps” to “How much it helps”

If the definition/meaning of $\ell^{H}$ was clarified and the typos were corrected, I would be prepared to raise my score.

**Ethical Concerns:**

["NO or VERY MINOR ethics concerns only"]

**Final Justification:**

My first impressions of the paper were already positive. My main concern was that some parts of the paper were difficult to read and understand, such as the initially mysterious $\ell^H$ quantity. The authors’ rebuttal has addressed my concerns about clarity, and my impression of the paper remains very positive. Therefore, I strongly feel that this paper should be accepted.

**Limitations:**

Yes.

**Paper Formatting Concerns:**

None.

**Quality:**

3

**Strengths And Weaknesses:**

**Strengths:**
The paper contains several nice ideas, such as the observation that a fixed-horizon betting algorithm should always finish with wealth of either 0 or $1/\delta$.

I like some aspects of how the ideas of the paper are presented. For instance, the examples (or scenarios) presented throughout the paper help to understand how an optimal fixed-horizon betting algorithm should behave.

The paper contains several nice results and proofs. For instance, I find Proposition 7 and its proof to be quite satisfying.

**Weaknesses:**
The main weakness is that some parts of the paper were very difficult to read and understand. In addition, there were many typos and strange sentences (see Questions for details).

A minor weakness is that the paper is does not quantify how much star-betting improves the theoretical bound.

---

> ### Author Rebuttal · Authors · 2025-07-28
>
> We thank the Reviewer for their careful reading of our work and for being available to raise their score if the definition/meaning of $\ell^{H}$ is clarified and the typos corrected.
> We will fix all the typos the Reviewers found and in the following we answer their specific questions.
>
> ## Questions
> **Confusion about $\ell^H$**\
> We are sorry for the (understandable) confusion, we will fix this. The set of feasible lambdas $C$ depends on $\sum_i X_i$ and it should have been a condition rather than a constraint set. $\ell^H$ is the argmin over $\lambda$ of the following optimization problem ($m,n,\delta$ are constants):
> $$
> \min_{X_1, \dots X_n \in [0,1], \lambda \in \mathbb{R}} \sum_{i=1}^n X_i, \quad
> \text{s.t.}\quad  \lambda\sum_{i=1}^n (X_i-m) -n\lambda^2/8 \geq \log\frac1\delta
> $$
>
>
> **How quickly does $c$ decay in theorem 8?**\
> It is easy to calculate that $c$ decays as $n^{-1/8}$ (this treats all the other problem parameters as constants when we only increase $n$) when selecting constants in the proof as In Appendix A (lines 405-408). That said, we did not try to optimize this rate and a potentially better choice of constants might exist. We will elaborate on this in the revised version of the paper.

---

> > ### Comment · Reviewer_QY1z · 2025-08-04
> >
> > Thank you for your rebuttal. The quantity $\ell^H$ and the value $\ell^H = \sqrt{8\log(1/\delta)/n}$ in Algorithm 1 make a lot more sense to me now. As promised, I’m happy to raise my score. I believe this paper should absolutely be accepted.

---

> > > ### Author Response · Authors · 2025-08-05
> > >
> > > We thank a lot the Reviewer for engaging with us and for increasing the score. Your appreciation of our work means a lot to us.

---

> > > > ### Author Response · Authors · 2025-08-08
> > > >
> > > > We thank again the Reviewer and we encourage them to ask further questions before the deadline if needed. If not, please do update your score as promised above

---

### Official Review · Reviewer_sGJY · 2025-07-11

**Clarity:** 3
**Significance:** 4
**Originality:** 3
**Rating:** 5
**Confidence:** 3

**Summary:**

This paper considers the setting of construction of  $(1-\delta)$CI of a bounded random variable from $n$ i.i.d. Samples for a given $\delta$. The main contribution of the paper is to develop a novel technique to improve the performance of existing betting based methods for construction of CI as well as improving the CI based on concentration bound such as Hoeffding and Bernstein. The improved betting algorithm, i.e., STaR performs better than the existing methods for CI construction including the state of the art betting based methods.

**Questions:**

Questions:

1- How does one show that the optimal width is of order of $O(\sqrt{\frac{\sigma^2 \log(1/\delta)}{n}})$?

2 In Equation 1, the confidence interval is defined such that the probability of error on both the left and right sides is $\delta/2$. Doesn’t this exclude cases where the error is not equally split, but the total error is still $\delta$?

Suggestions:

1- If I understand it correctly, I think there is a typo in the equation given below line 119.

**Ethical Concerns:**

["NO or VERY MINOR ethics concerns only"]

**Final Justification:**

I thank the authors for their clarification. I think, they have answered my questions appropriately and I am raising the score.

**Limitations:**

Yes

**Quality:**

3

**Strengths And Weaknesses:**

Strengths:

1- Idea of improving the betting based methods in novel and Intuition of why STaR algorithm works well is explained well through an example in Section 3.

2- Numerical experiments are quite interesting. I really liked the comparison with Clopper-Pearson method in Figure 3.


Weakness:

1-  Theorem 3 is only provided for Algorithm 3.  Maybe I am missing something but how does that lead to theoretical support for Algorithm 4?

2- The paper is only written for bounded random variables. This somewhat restricts the scope of the paper. I wonder if this technique can be useful in more general settings.

---

> ### Author Rebuttal · Authors · 2025-07-28
>
> We thank the Reviewer for their feedback. In the following we address their comments and questions.
>
> ##  Weaknesses
> **Theorem 3 (typo, should be thm 8) only considers alg 3.**\
> Yes, this is true (and clearly stated in the statement). We have proven that STaR technique helps certain classes of betting strategies, but the improvements of Alg. 4 over Alg. 3 are only empirical. That said, Appendix C provides evidence that already Alg. 3 beats the SotA empirically, and in terms of guarantees by virtue of Theorem 8.
>
> **The focus is on bounded random variables - what about more general settings?**\
> Generally speaking, the technique is not limited to bounded random variables but to betting algorithms of a specific form. Given how fundamental is the setting of bounded random variables, both in theory and in practice, we find it very difficult to consider it as a limitation. As a proof of this point of view, the most important work in this area (Waudby-Smith and Ramdas 2024) considers exactly the same setting. As a matter of fact, we started working on this topic because applied people explicitly asked us for a better method for the calculation of valid confidence intervals of bounded random variables for deep learning applications, as, for example, in
>
> Angelopoulos AN, Bates S, Fannjiang C, Jordan MI, Zrnic T, Prediction-powered inference, Science, 2023
>
> In future work, we will explore the application of the STaR method to other settings in the testing-by-betting literature.
>
> ## Questions
> **How is  $O(\sqrt{\frac{\sigma^2 \log(1/\delta)}{n}})$ the right order?**\
> Intuitively, this is predicted by central limit theorem, where the sample mean $\frac{1}{n} \sum_{i=1}^{n} X_i$ behaves like a normal random variable with standard deviation $s = \sqrt{\sigma^2/n}$, then the standard Gaussian tail bound asserts $ \delta = P[ X - E[X] > t] \leq \exp(-t^2/(2s^2))$, then solving for $t$ gives us the desired expression (we have that $\sigma^2 = \text{Var}[X]$).
>
> **Combining lower and upper confidence bound at different confidence levels**\
> Yes, this is completely possible. In (1) we presented the standard way to get two-sided confidence intervals from the one sided ones (the focus of the paper are the one sided ones). If the application requires splitting the probabilities differently, everything will work just as good.
>
> **Typo after line 119**\
> Yes, thank you, there should have been a minus sign in the second bracket.

---

> > ### Comment · Reviewer_sGJY · 2025-08-06
> > **Follow up**
> >
> > Thank you, authors, for your response; it does address my questions. Following up on the discussion regarding the optimal width order analysis for betting-based methods, I have a follow-up question concerning your claim in the abstract (lines 10–11):
> >
> > No betting-based algorithm guaranteeing the optimal
> > $$
> > \sqrt{\frac{\sigma^2 \log(1/\delta)}{n}}
> > $$
> > width of the confidence intervals is known. In my opinion, this result can be derived from Theorem 4.6 of [1]. Moreover, [2] have shown that a Taylor series expansion of $\mathrm{KL}_{\inf}(P,x)$ in $x$
> > is possible near the true mean of $P$ (see Lemma 2 in Section 7 of [2]), and that its second derivative at the mean equals $1/\sigma^2$. Therefore, one should be able to obtain the
> > $$
> > \sqrt{\frac{\sigma^2 \log(1/\delta)}{n}}
> > $$
> > width for the betting-based method from Theorem 4.6 of [1].
> >
> > Could you please clarify on this?
> >
> > Reference
> >
> > [1] - Shekhar, Shubhanshu, and Aaditya Ramdas. "On the near-optimality of betting confidence sets for bounded means." arXiv preprint arXiv:2310.01547 (2023).
> >
> >
> > [2]- Deep, Vikas, Achal Bassamboo, and Sandeep Juneja. "Asymptotically optimal adaptive a/b tests for average treatment effect." Available at SSRN 4642974 (2023).

---

> > > ### Author Response · Authors · 2025-08-06
> > >
> > > **No betting-based algorithm guaranteeing the optimal width of the confidence intervals is known: Could you clarify it?**
> > >
> > > Theorem 4.6 in [1] *does not* achieve the optimal bound due to a spurious $\log(n)$ factor. This issue is discussed at the beginning of Section 4.2 in [1]. The $\log(n)$ factor is due to using a mixture betting strategy. Mixture strategies can produce optimal confidence sequences (i.e., the LIL bounds), but nobody has yet found a way to change the mixture to produce truly optimal confidence intervals (i.e., for finite $n$). As far as you know, it might be impossible. This shortcoming is well known in this community and it is exactly why we focused on a different betting approach, not based on a mixture.

---

> > > > ### Comment · Reviewer_sGJY · 2025-08-09
> > > >
> > > > Thank you for the response. Based on the discussion, I am raising the score now.

---

> > > > > ### Author Response · Authors · 2025-08-09
> > > > >
> > > > > Thank you!

---

### Author Response · Authors · 2025-08-08

Dear Reviewers (specially those who has not responded yet) and AC,

The discussion deadline is approaching, and delaying your response will reduce our chance to interact. We’d greatly appreciate your reply as soon as possible.

---

### Note · Authors · 2025-08-12

Dear AC and Reviewers,

All reviewers noted that our algorithm outperforms the state of the art for the calculation of valid confidence intervals of a bounded random variable ---- a canonical statistical estimation task --- and at least two reviewers support acceptance.

Reviewer Epqr mainly suggested applying our methods in e-value–related contexts, which we find interesting but orthogonal to our aims and outside the scope of our paper.

The two Reviewers with rejecting scores did not engage in any discussion with us (one only ticked the mandatory acknowledgment after the deadline for the discussion). Once again, we respectfully disagree that these two reviews support rejection and kindly ask the AC to carefully review them and our responses.

---

### Decision · Program_Chairs · 2025-09-17

**Decision:**

Accept (poster)

**Comment:**

The paper considers the problem of constructing confidence intervals for the mean of a bounded random variable, given access to a fixed number of samples, and proposes new methods for constructing confidence intervals using betting algorithms. Previous betting-based confidence intervals are either designed for anytime-validity, and as a result are sub-optimal for a fixed sample size, or lack non-asymptotic bounds on their width. The main idea behind the proposed star-betting method, is to design betting algorithms that always finish with either 0 wealth or just enough wealth to reject a candidate value for the mean, as opposed to betting algorithms that simply try to achieve maximal wealth.

The reviews were mixed but relying on my own expertise I can confidently recommend acceptance.